# A de novo missense variant in MIDEAS results in increased deacetylase activity of the MiDAC HDAC complex causing a neurodevelopmental syndrome

Louise Fairall [1,6], Kristupas Sirvydis [1,6], Robert E. Turnbull [1,6], Suzan JG Knottnerus[2,6], Oksana Gonchar[1], Frederick W. Muskett [1], Rebekah Jukes-Jones[1], Lonneke van Brussel[2], Ellen van de Geer[2], Koen van Gassen[2], Paul Badenhorst [3], Diana Johnson[4], Paulien A. Terhal[2], Peter M. van Hasselt[5], Richard H. van Jaarsveld [2] ✉ & John WR Schwabe [1] ✉

MIDEAS is a scaffold protein that, together with DNTTIP1, mediates assembly of the MiDAC histone deacetylase complex. Mice lacking MiDAC die before birth suggesting a key developmental function. Here, we report two unrelated individuals, with a multisystem disorder characterised by delayed speech development, joint contractures, dysmorphic features and dysmotility of the gut. Both individuals have the same de novo heterozygous missense variant in *MIDEAS* (p.Tyr654Ser). A cryoEM structure of the MiDAC complex reveals that this amino acid is located in a conserved auto-inhibitory loop that covers the active site of the deacetylase enzyme. We suggest that the variant results in loop displacement leading to elevated deacetylase activity. In support, we observe reciprocal gene expression changes in patient fibroblasts compared with a cell line following rapid MiDAC degradation. Our results establish *MIDEAS* as a dominant monogenic disease gene and that hyperactivity of the MiDAC complex results in a characteristic multisystem disorder.

Appropriate gene regulation is essential for controlling developmental processes as well as enabling adult organisms to respond to both internal and external challenges including metabolism and disease. Class I histone deacetylase enzymes HDACs 1, 2 & 3 are critical to the development and differentiation of all eukaryotes[1]. They are implicated in numerous nuclear processes, including transcription, splicing, DNA replication and repair, regulation of the cell cycle, cellular proliferation and X-chromosome inactivation. HDACs 1 and 2 are highly similar paralogues that are assembled interchangeably into six distinct multiprotein complexes: NuRD, CoREST, SIN3, MiDAC, RERE and MIER[2–7]. HDAC3 is assembled into the SMRT/NCoR complexes[8]. These HDAC complexes differ in subunit composition, often containing chromatin and transcription factor interaction activities. The different subunit compositions determine the function and specificity of the different complexes. Mice lacking specific components of these complexes typically die during embryogenesis, suggesting little or no redundancy[5,9–11]. The clinical relevance of these HDAC complexes is highlighted by more than 100 active clinical trials of HDAC inhibitors (clinicaltrials.gov).

[1]Institute for Structural and Chemical Biology, Department of Molecular and Cell Biology, University of Leicester, Leicester, UK. [2]Department of Genetics, University Medical Center Utrecht, Utrecht, The Netherlands. [3]Institute of Cancer and Genomic Sciences, University of Birmingham, Edgbaston, UK. [4]Department of Clinical Genetics, Sheffield Children's NHS Foundation Trust, Sheffield, UK. [5]Department of Metabolic Disease, Wilhelmina Children's Hospital, University Medical Centre Utrecht, Utrecht, the Netherlands. [6]These authors contributed equally: Louise Fairall, Kristupas Sirvydis, Robert E. Turnbull, Suzan JG Knottnerus. ✉e-mail: R.H.vanJaarsveld@umcutrecht.nl; john.schwabe@leicester.ac.uk

The MiDAC complex exists as both dimers and tetramers and contains three proteins: MIDEAS, HDAC1 and DNTTIP1[5,12]. Although the MiDAC complex represents less than 5% of the class I HDAC complexes in the cell[13], it has been implicated as having a role in mitosis[14] and several reports show interaction with CDK2/CyclinA[15]. More recently, it has been shown that MiDAC is important for chromosome alignment during mitosis in cancer cell lines[5]. Mice lacking either *Dnttip1* or *Mideas* die during late embryogenesis with heart malformation and haematopoietic failure[5]. MiDAC has also been shown to play a crucial role in regulating signalling through SLIT3 and NTN1 during neurite development[16].

Genes involved in epigenetic regulation are increasingly recognised as monogenic disease genes[17–19], often associated with neurodevelopmental disorders. This class of neurodevelopmental disorders is generally characterised by intellectual disability and/or abnormal growth[20]. Whereas several members of HDAC complexes have been identified as monogenic disease genes, none of the MiDAC components (DNTTIP1, HDAC1 and MIDEAS) have been reported to cause Mendelian disease.

Here, we report two unrelated individuals with overlapping clinical features, including speech delay, joint contractures, dysmorphic features and gastrointestinal motility problems. Both individuals harbour a de novo heterozygous missense variant, p.Tyr654Ser (Y654S), in a conserved part of the MIDEAS protein adjacent to the ELM2-SANT domain. A 2.9 Å cryo-EM structure of the MiDAC dimer reveals that Y654 is located in an auto-inhibitory loop that lies on the surface of HDAC1 and likely regulates the activity of the enzyme. The substitution from YTP to STP generates a CDK kinase consensus site that is phosphorylated in human cells. The Y654S complex exhibits increased deacetylase activity in vitro against histone peptide substrates, suggesting a hyperactive complex and that the loss of auto-inhibition contributes to pathology.

Comparison of gene expression profiles from mouse fibroblasts and a human colon cancer cell line lacking the MiDAC complex with fibroblasts derived from one of the probands shows reciprocal changes in gene expression, further supporting the hypothesis that the Y654S variant is hyperactive. Depletion of MiDAC in HCT116 cells results in upregulation of MAP2K6 and MAP2K3, suggesting that one role for MiDAC may be to act as a regulatory brake on the p38 MAPK pathway. This may partly explain why hyperactive MiDAC causes a dominant neurodevelopmental syndrome. These results establish *MIDEAS* as an autosomal dominant monogenic disease gene, which causes a neurodevelopmental disorder, most likely through hyperactivity of the MiDAC complex.

## Results

### Identification of a de novo *MIDEAS* variant in two unrelated probands

We performed clinical trio-exome sequencing for Proband 1, an individual suspected of having a monogenic neurodevelopmental disorder, born to healthy parents with no relevant family history. We considered variants according to 1) de novo or recessive inheritance, and 2) the ACMG/AMP guidelines for variant interpretation[21]. No variants were identified that could explain the phenotypes within known disease genes. In addition, transcriptome-wide expression analysis on patient-derived fibroblasts revealed no suspect expression outliers (Supplementary Fig. 1a). We therefore looked for genetic variants outside of known disease genes and found one potential candidate: a de novo heterozygous missense variant in *MIDEAS* (NM_001367710.1: c.1961A>C (p.Tyr654Ser); hereafter referred to as Y654S) (Fig. 1a–c).

The variant was absent from the Genome Aggregation Database[22] and located in a region with little variation in the general population (Fig. 1b, c). The affected residue is conserved in nematodes, drosophila, mouse and humans and is located within an 18 amino acid region of conservation (aa: 653-670, shaded blue in Fig. 1c). The variant is

predicted to be damaging by both AlphaMissense and Primate AI. Interestingly, there is an additional conserved region just upstream (aa: 631-637, shaded yellow in Fig. 1c) that also shows very little variation. Based on the conservation of the Y654 residue and its immediate surroundings, both within humans and across species, we considered Y654S a promising candidate.

We therefore searched for additional cases with *MIDEAS* variants using the online matchmaking platform Genematcher[23] and found one additional case. Strikingly, this individual (Proband 2) had the same de novo variant and the two probands presented with overlapping clinical features (Table 1, Fig. 1d, e and Supplementary Table 1, Supplementary Note 1).

Proband 1 was 26 and Proband 2 was 9 years old at time of last clinical examination. Both presented with global developmental delay, feeding difficulties, velopharyngeal insufficiency and hearing loss. In addition, both probands presented with similar craniofacial dysmorphisms, including ptosis, narrow palpebral fissures, limited facial expression and a small mouth (Fig. 1d, e). Notably, both have generalised joint contractures which seem to be especially progressive during infancy (Fig. 1d, e). They both exhibit gastro-intestinal symptoms, with infantile onset chronic diarrhoea. In Proband 1, the gastrointestinal problems led to the development of a chronic idiopathic intestinal pseudo-obstruction.

Proband 1 additionally has a pronounced growth delay and a disproportionate stature with relatively short limbs. Thus far, growth is not clearly affected in Proband 2 (height −1.35 SD at the age of 9 years). Motivated by the identification of an identical de novo variant in two unrelated probands, who presented with striking phenotypic overlap, we set out to understand the molecular impact of Y654S.

### Gene expression changes in patient-derived fibroblasts

The MIDEAS protein is a core component of the MiDAC histone deacetylase complex that plays a role in gene regulation[5,16]. We therefore used clinical RNAseq data from Proband 1 to determine a set of differentially expressed genes (zScore > 2) in patient dermal fibroblasts compared with 30 other diagnostic fibroblast samples. There are 132 upregulated and 303 downregulated genes after mapping to known ensembl gene IDs (Supplementary Fig 1a). Gene ontology for both "biological processes" and "molecular function" identified multiple pathways that are perturbed by the *MIDEAS* variant (Fig. 2a).

To explore whether the variant perturbs MiDAC complex function, we compared the gene expression profile in Proband 1-derived fibroblasts with a previously established experimental knockout model in mouse embryonic fibroblasts[5]. Interestingly, of the 420 genes that are perturbed in the patient, 177 are observed in the mouse knockout model fibroblasts, although there are rather few genes that are significantly perturbed (Fig.2b–d and Supplementary Fig. 1b). Overall, there is a negative correlation of −0.21 indicating a partial reciprocal relationship.

### Cryo-EM of the MIDEAS dimer reveals an autoinhibitory loop

We have previously solved the structures of the MiDAC dimer and tetramer to 4.0 and 4.5 Å, respectively[5]. These structures include full-length HDAC1, the dimerisation domain from DNTTIP1 (aa: 1–130) and the ELM2-SANT domain from MIDEAS (aa: 717–887). The structures of the MIDEAS protein did not contain the 18 amino acid conserved sequence in which the Y654S variant is located (aa: 653–670, shaded blue in Fig. 1c) nor did they contain the additional upstream conserved region (aa: 631–637, shaded yellow in Fig. 1c).

To investigate the structure of the MiDAC complex containing a longer construct of MIDEAS, we expressed MIDEAS (628-887), which includes the two conserved sequences, together with full-length DNTTIP1 (including the DNA-binding domain) and full-length HDAC1 (Supplementary Fig. 2a). We then prepared a higher-order complex bound to di-nucleosomes purified from HEK293F cell nuclei and

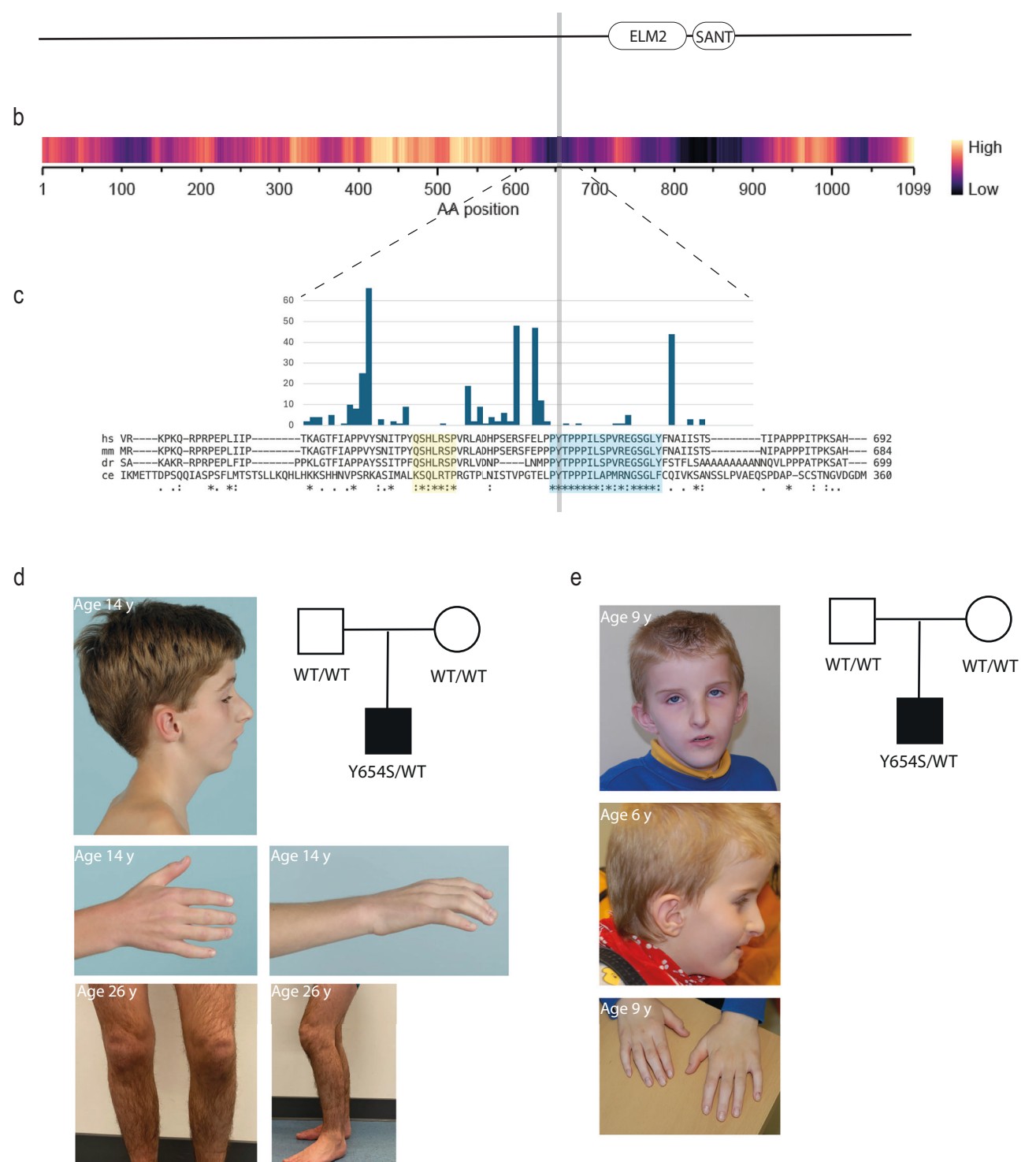

**Fig. 1 | The human MIDEAS variant Y654S. a** Schematic of the MIDEAS protein showing location of the ELM2 and SANT domains. **b** Density of missense variants across *MIDEAS* according to gnomAD V4.1.0. High density means more variants reported in that region. **c** Zoom-in on region surrounding Y654S, showing occurrence of gnomAD missenses (bar graph) and residue conservation across species. *Homo sapiens* MIDEAS (hs), *Mus musculus* MIDEAS (mm), *Danio rerio* MIDEAS (dr) and *Caenorhabditis elegans* SAEG-1 (ce). **d** and **e** Pictures depicting facial dysmorphisms and joint contractures, and pedigrees with respective genotypes for Probands 1 & 2, respectively.

fractionated using a Superose 6 size exclusion column (Supplementary Fig. 2b, c). Electromobility shift assays showed >90% binding with equimolar di-nucleosome and MiDAC complex at micromolar concentrations (Supplementary Fig. 2d).

A cryo-EM dataset of this complex enabled us to determine a significantly higher resolution (~2.9 Å) of a MiDAC dimer complex compared with our previous structure (Fig. 3a–i and Supplementary Fig. 2e, 3–5). Unfortunately, due to the flexibility of the complex, we

**Table 1 | Clinical features in two probands with the c.1961A>C [p.Tyr654Ser] MIDEAS variant**

| | Proband 1 | Proband 2 |
|---|---|---|
| Age at last examination (years) | 26 | 9 |
| MIDEAS variant cDNA (GenBank: NM_001367710.1) | c.1961A>C | c.1961A>C |
| Polypeptide | p.Tyr654Ser (p.Y654S) | p.Tyr654Ser (p.Y654S) |
| Genomic DNA GRCh38 (hg38): chr14:73729774 T > G | chr14:73729774 T > G | chr14:73729774 T > G |
| Inheritance | De novo | De novo |
| Onset | ~4 months Developmental delay, failure to thrive | Birth, respiratory distress, dysmorphic features noted |
| Craniofacial dysmorphism | Ptosis Short and narrow palpebral fissures Low-hanging columella Limited facial expression Narrow mouth Prominent nasal bridge | Ptosis Short and narrow palpebral fissures Low-hanging columella Limited facial expression Narrow mouth |
| Birth weight | Normal | Normal |
| Growth | Postnatal growth failure with adult short-limb disproportionate stature | Not applicable |
| Height at last examination | 140 cm (−6.18 SD) | −1.35 SD |
| Development | Delayed motor and speech development, SON-R 71 at age nine years, SON-RS 84, SON-PS 63 | Delayed development of speech at age 6 years, autism |
| Muscoloskeletal features | Multiple joint contractures, very stiff joints, slightly progressive | Contractures knees |
| Gastrointestinal | Diarrhoea Volvulus Gastrointestinal dysmotility with pseudo-obstruction | Diarrhoea Nasogastric tube from age 8 months PEG fed from 20 months to 7 years |
| Hearing/vision | Mixed (mainly sensorineural) hearing impairment | Conductive hearing impairment |
| Integument | Thickened skin Eczema | Eczema |

are unable to see either the DNA-binding domains of DNTTIP1 or the interaction of MiDAC with a nucleosome. However, the resulting 2.9 Å map shows new features that were not seen in the previous MiDAC dimer structure (Supplementary Fig. 5) including residues 35–61 of the N-terminus of DNTTIP1. Within this region residues P37-M41 are interleaved between the ELM2 and SANT domains of MIDEAS making extensive interactions with conserved amino acids in both domains (Fig. 3c, d). In addition, residues H44-F57 form an ordered loop that protrudes from the structure and mediates several key interactions with HDAC1 (Fig. 3e).

The most important insights from the new structure concern the additional amino acids N-terminal to the ELM2-SANT domain of MIDEAS (aa: 628–716). Both conserved regions (aa: 631–637 and 653–670) are clearly visible in the new map and reveal MIDEAS completely wrapping around HDAC1 and unexpectedly covering the active site of the enzyme (Fig. 3f, g). Importantly, I659 from MIDEAS is positioned in the active site channel of HDAC1, making interactions with two phenylalanines in HDAC1 (F150 and F205). The location of this loop strongly suggests that it acts either to inhibit or modulate the HDAC activity. Y654, the residue affected in the probands, makes a hydrogen bond to the backbone carbonyl of P206 HDAC1. The aromatic ring is stacked between two prolines (P721 and P656) in MIDEAS (Fig. 3h). These prolines are conserved in the human paralogues of MIDEAS (TRERF1 and ZNF541), as well as homologues, including the *C. elegans* SAEG-1. The serine residue in the variants would not make as favourable an interaction with P656 and P271 and would likely be too far to make a hydrogen bond with HDAC1.

Strikingly, the conserved amino acid stretch 631–637 upstream of the auto-inhibitory loop (shaded yellow in Fig. 4a) adopts an extended conformation that lies directly over the top of the auto-inhibitory loop, making key interactions with HDAC1 on either side. It is likely that this secondary loop serves to stabilise the proposed auto-inhibitory loop over the active site of HDAC1 (Fig. 4b).

## Y654S creates a CDK consensus phosphorylation site

Mutation of Y654 to a serine residue would likely perturb the interaction of the auto-inhibitory loop with HDAC1 due to loss of favourable interactions. It would also result in an STP peptide that, in contrast to YTP, is a substrate for Cyclin Dependent Kinase (CDK) phosphorylation on the threonine[24]. Phosphorylation of T655 would likely be particularly damaging to the interaction of the auto-inhibitory loop since T655 makes a hydrogen to E146 in HDAC1. Phosphorylation would not only break this hydrogen bond but would lead to charge repulsion (Fig. 3i).

Since the MiDAC complexes are purified from HEK293F cells, which express normal levels of cyclins and CDKs, we used mass spectrometry to determine the post-translational modifications (PTM) in the wild-type (WT) and Y654S mutant MIDEAS protein. No peptides were detected with phosphorylation of T655 in the WT (YTP) protein, however phosphorylation was detected on more than 41% of the peptides containing STP either on S654 or T655 (Fig. 4c). Just 2 h treatment with a CDK2/9 inhibitor, flavopirol hydrochloride, after 2 days expression of the Y654S MiDAC complex, results in a 22% reduction in phosphorylation, supporting the hypothesis that CDKs may be responsible for this phosphorylation (Fig. 4c). This fits with the reported interaction of the MiDAC complex with the CDK2:CyclinA2 complex[15].

## Deacetylase activity of WT and mutant MIDEAS dimer

To investigate whether the proposed auto-inhibitory loop does indeed cause inhibition of HDAC activity, we employed an NMR-based assay, using an acetylated histone H3K9 peptide, to determine the activity of the MiDAC complex with and without the MIDEAS auto-inhibitory loop (Fig. 4d, e). We observed a marked difference in deacetylase activity. The complex containing a shorter MIDEAS protein (aa: 717–887), lacking the auto-inhibitory loop, showed an initial deacetylation rate of 29.7 nM.s$^{-1}$. In striking contrast, the complex with the longer MIDEAS protein (aa: 628–887) was five times less active (initial rate 6.3 nM.s$^{-1}$). The complex containing the longer MIDEAS protein bearing the Y654S

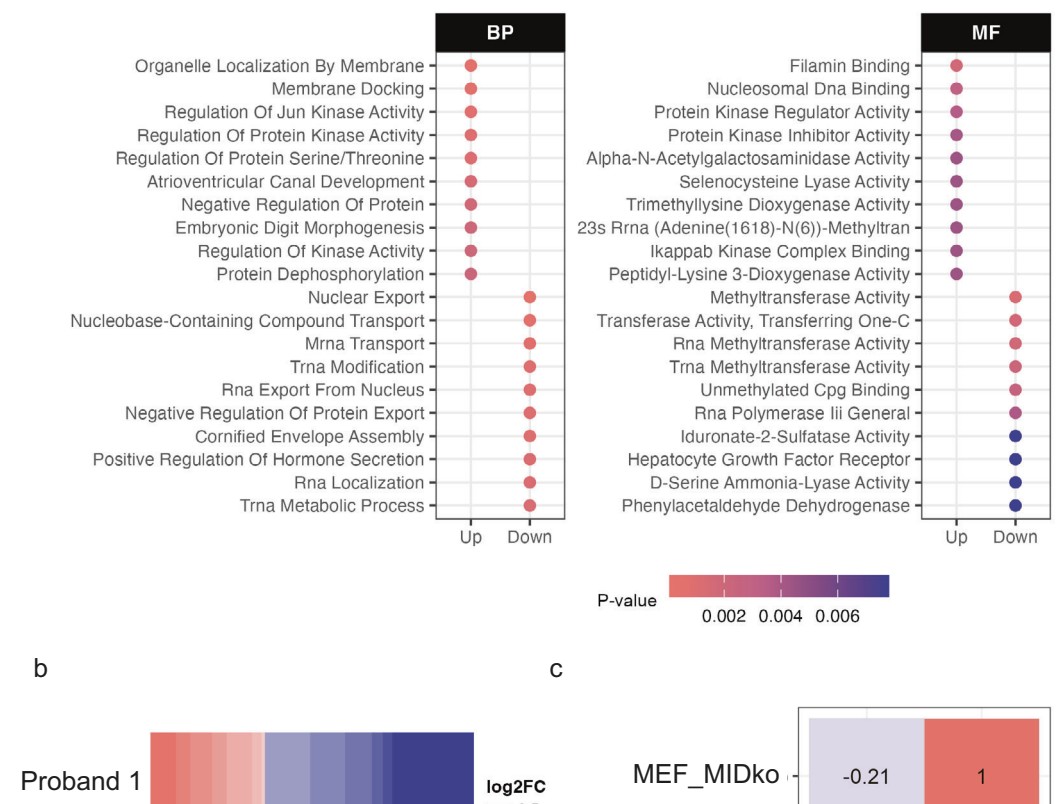

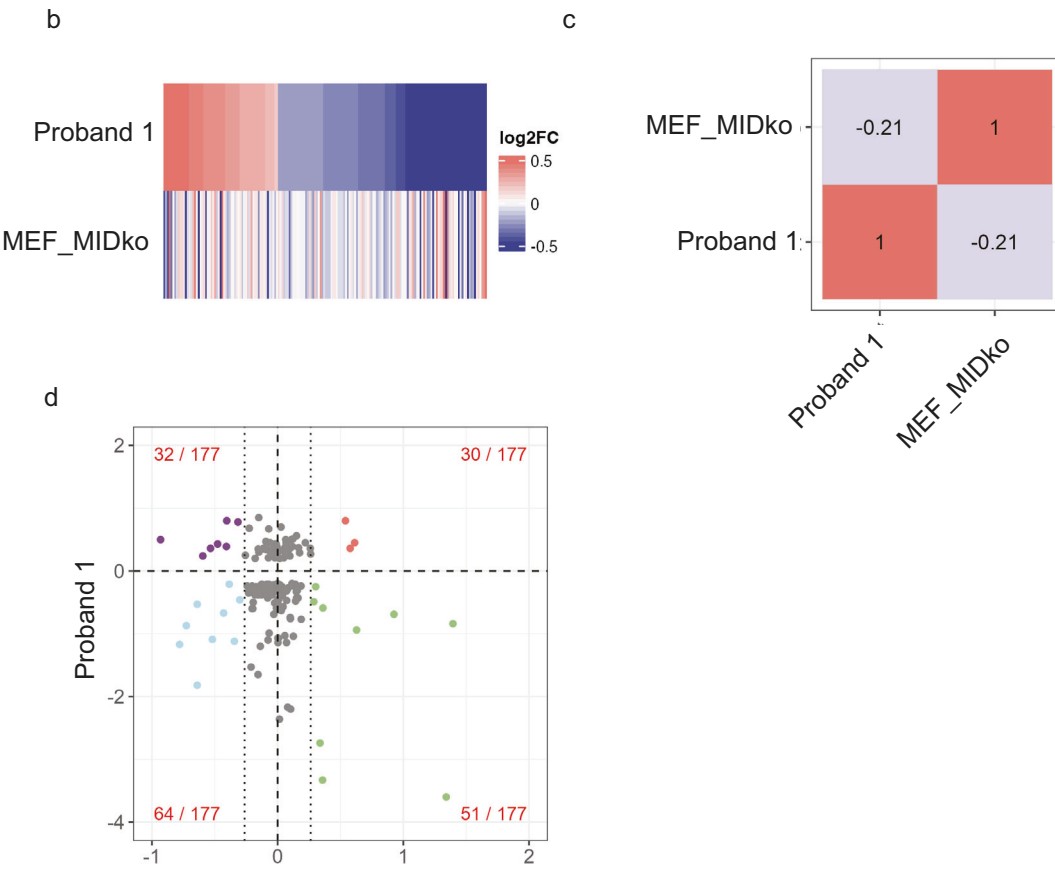

**Fig. 2 | Comparison of gene expression changes between Proband 1 and knockout MEFs. a** Gene ontology analysis for biological processes (BP) and molecular functions (MF) on Proband 1 up and down-regulated genes (zScore > 2) identified by RNAseq. Significant pathways were calculated using a classic one-sided Fisher exact test with a cut-off <0.01. The top 10 pathways are plotted. **b** Heatmap comparing the log2 fold change of Proband 1 genes (zScore >2) and the same genes in the knockout MIDEAS mouse embryonic fibroblast dataset (fold change limit set to -0.5 – 0.5 to aid visualisation). **c** Overall correlation of fold changes of Proband 1 genes (zScore >2) vs the same genes in the knockout MIDEAS mouse embryonic fibroblast dataset (MEF_MIDko). **d** Fold change of overlapping genes in both Proband 1 and MIDEAS KO MEF datasets.

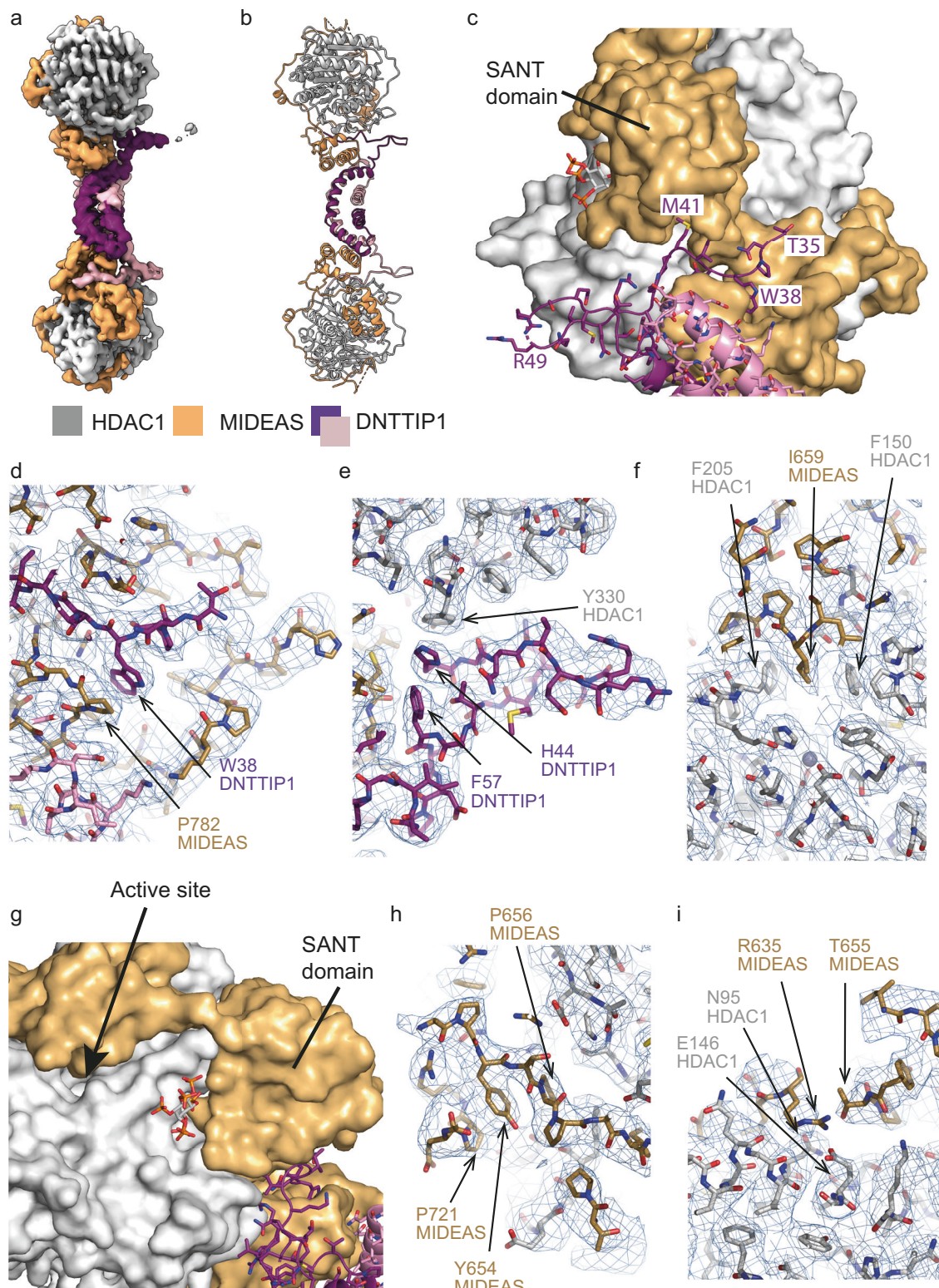

**Fig. 3 | Structure of the MiDAC dimer to 2.9 Å. a,b** View of the overall combined focussed map with a cartoon representation of the MiDAC complex. HDAC1 is shown in grey, MIDEAS orange, DNTTIP1 in purple and pink. Map contour level is 10 in Chimera. **c** The N-terminal tail of DNTTIP1 (shown as a purple cartoon and sticks) forms a loop and interacts in a groove in MIDEAS (shown as an orange surface). **d** Interaction of DNTTIP1 T35 to M41 with the MIDEAS ELM2 and SANT domains (shown in the map). **e** N-terminal loop of DNTTIP1 from H44 to H57 (shown in the map). **f** Interaction of MIDEAS P653 to Y670 with HDAC1 showing the potential inhibition of HDAC1 by I659 in the active site (shown in the map). **g** The active site of HDAC1 (grey) highlighted by an arrow is sterically hindered by MIDEAS. MIDEAS and HDAC1 are shown as a surface and DNTTIP1 is shown in sticks. **h** Interaction of MIDEAS P653 to Y670 with HDAC1 showing the interaction of Y654 MIDEAS with P721 and P656 (shown in the map). **i** T655 of MIDEAS interacts with N95 and E146 of HDAC1 (shown in the map). Map contour level is 9 in Pymol.

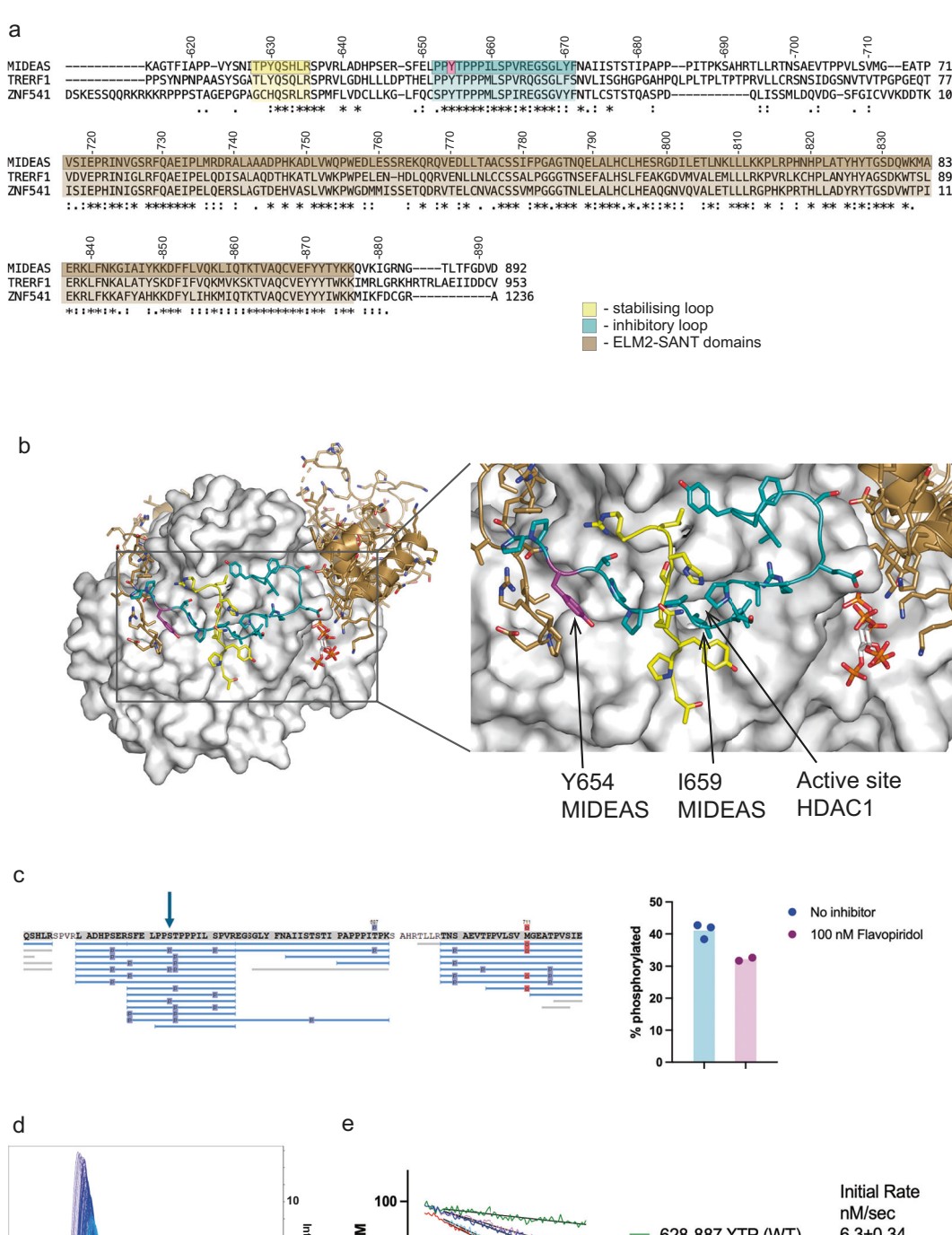

**Fig. 4 | Inhibitory loop of MIDEAS. a** Alignment of the homologous proteins TRERF1, MIDEAS and ZNF541 to show the conserved regions N-terminal to the ELM2-SANT domains. T628-R635 is shown in yellow. P652-F671 is shown in cyan. The ELM2-SANT domain is shown in brown. Y654 is shown in magenta. **b** MIDEAS (coloured based on the alignment) shown as a cartoon and sticks completely wraps around HDAC1 (grey) shown as a surface with a closeup of the conserved N-terminal regions of MIDEAS. **c** Graphical representation of mass spectrometry analysis of peptides from Y654S MIDEAS and a bar chart showing the level of phosphorylation of either the serine or threonine in the STP sequence with (n = 2) and without (n = 3) the kinase inhibitor flavopiridol. Phosphorylation sites are shown as P in a purple box. STP is highlighted with a blue arrow. **d** Peaks obtained by deacetylation of a H3K9ac peptide in a time course over 2000 secs coloured blue to violet. **e** Initial rates of reaction calculated for MIDEAS complex with MIDEAS 628-887(YTP) WT, MIDEAS 717-887 (WT), MIDEAS 628-887 Y654S (STP), MIDEAS 628-887 Y654S T655A (SAP) and MIDEAS 628-887 Y654S T655D (SDP). Source data are provided as a Source Data file.

mutation (STP in place of YTP and presumably bearing >34% phosphorylation on the S654/T665) had an initial rate of 20.4 nM.s$^{-1}$. A mutant complex bearing SAP (not a CDK substrate) was somewhat less active (16.6 nM.s$^{-1}$). In contrast, a mutant complex bearing the phosphorylation mimic SDP was almost as active (27.3 nM.s$^{-1}$) as the complex lacking the auto-inhibitory loop (Fig. 4d, e).

These findings support the hypothesis that the conserved amino acids amino-terminal to the ELM2-SANT domain in MIDEAS serve an auto-inhibitory function, which is largely lost in the human variant, Y654S, leading to a complex that is 3–5 times more active than the WT complex.

### The MiDAC complex may act as a regulatory brake on the p38 MAPK cascade

To support our hypothesis that Y654S results in hyperactivity of the MiDAC complex, we reasoned that depleting the MiDAC complex would result in reciprocal effects on gene expression. Since the MiDAC complex has previously been shown to be dependent upon DNTTIP1[5,16,25], we used gene editing to introduce an FKBP12$^{F36V}$ tag into both alleles of the endogenous DNTTIP1 gene in the human HCT116 cell line. Treatment with the dTAG$^V$-1 PROTAC[26] induced rapid degradation of DNTTIP1 in less than one hour (Supplementary Fig. 7a). It is likely that the functional MiDAC complex is lost, since DNTTIP1 forms the oligomerisation core of MiDAC, and co-dependency of MiDAC components has been described previously[5,16]. Bulk RNAseq at various timepoints after dTAG$^V$-1 treatment showed surprisingly few significant changes in gene expression, with the largest fold-change increase in *MAP2K6* (Fig. 5a, b and Supplementary Fig. 7b), although normalised counts following treatment were the lowest compared to other proteins in the pathway (Fig. 5b). Changes in *MAP2K6* expression were also observed using nascent qPRO sequencing, demonstrating active upregulation of *MAP2K6* transcription, and in our previous MEF *MIDEAS* and *DNTTIP1* KO data (Fig. 5c, Supplementary Fig. 7c, d)[5]. Interestingly, a more modest increase in the expression of *MAP2K3* was also consistently observed across model systems (Fig. 5a, b and Supplementary Fig. 7c, e). Importantly, western blots from treated and untreated HCT116 cells show that *DNTTIP1* deletion results in a significant increase in MAP2K6 protein levels as well (Fig. 5d). Taken together, this suggests that the MiDAC complex may act as a regulatory brake on the p38 MAPK cascade. Additionally, Mondal et al. and Wang et al. have observed using ChIP-seq that the MiDAC complex associates with both *MAP2K6* and *MAP2K3* genes and is lost following knockout of either MIDEAS or DNTTIP1 with a corresponding, modest, increase in acetylation (Supplementary Fig. 8)[16,25].

Comparison of the gene set from Proband 1 fibroblasts, with gene expression changes in HCT116 cells following dTAG$^V$-1 treatment, showed a striking reciprocal relationship (negative correlation, represented by negative values) that becomes stronger with longer treatment with dTAG$^V$-1 (Fig. 5e). This supports the hypothesis that the Y654S variant leads to a hyperactive MiDAC. Interestingly, it also suggests that the downstream gene expression changes are the result of long-term perturbation of the MiDAC complex signalling. Mondal et al. and Wang et al. also observed MiDAC complex associated with many of these reciprocally regulated genes (Supplementary Figs. 9, 10)[16,25]. Again, this association is lost following knockout of either MIDEAS or DNTTIP1 which leads to a modest increase in acetylation (Supplementary Figs. 9, 10)[16,25].

## Discussion

Here we present two unrelated probands with the same de novo MIDEAS variant (NM_001367710.1: c.1961A>C (p.Tyr654Ser)) and overlapping phenotypic features, revealing that *MIDEAS* is an autosomal dominant monogenic neurodevelopmental disorder gene.

A 2.9 Å cryoEM structure shows that Y654 is located in a conserved loop that wraps directly over the active site of the HDAC1,

positioning an isoleucine side chain in the active site channel. The structure suggests that the Y654S mutation would displace this loop, increasing accessibility of the HDAC1 active site. Supporting this, functional studies show that this loop inhibits the deacetylase activity and that the Y654S mutation does indeed result in increased HDAC activity. The dissociation of the auto-inhibitory loop may be enhanced through the mutation creating a CDK site that we show is highly phosphorylated in a human cell line. Intriguingly, components of the MiDAC complex have previously been shown to interact with both CDK2 and CyclinA2 in G1S phase of the cell-cycle suggesting that the MIDEAS variant may be a proximal target of CDK2 and may be influenced by, or play a role in, the cell cycle[15].

The reciprocal relationship between expression profiles in patient fibroblasts and our engineered loss-of-function models confirms that Y654S activates the MiDAC complex supporting the suggestion that the aetiology of the disorder is inappropriate hyperactivity of the histone deacetylase in the MiDAC complex.

We show that depletion of the MiDAC complex in a cancer cell line results in upregulation of the kinases MAP2K6 and MAP2K3. They are both part of a signalling pathway that leads to phosphorylation of p38 (aka MAPK14). The pathway responds to numerous TGFβ signalling stimuli, as well as a variety of stress-related signals. The relationship of MIDEAS and DNTTIP1 with p38 is further supported by DEPMAP, which shows reciprocal co-dependencies (Pearson correlations of co-dependency: 0.25 and 0.19, respectively) which are in the top five hits for each protein[27]. As MiDAC acts to repress MAP2K6 (Fig. 5c), we hypothesise hyperactivity of the complex through Y654S only impacts on MAP2K6 function under conditions where the gene is activated, such as stress-related TGFβ stimuli.

Interestingly, the *MIDEAS*-related phenotype shows striking overlap with Myhre syndrome (OMIM 139210), including developmental delay, deep set eyes, palatal abnormalities, short stature, congenital heart defects, mixed hearing loss, brachydactyly with progressive joint limitations, thickening of the skin and gastrointestinal problems[28–30]. Myhre syndrome is caused by recurrent missense variants in *SMAD4*, which causes dysregulation of TGFβ signalling[28,31].

The phenotypic overlap with Myhre syndrome, and the observation that MiDAC regulates important TGFβ targets, suggests that dysregulation of the TGFβ pathway may contribute to the disorder resulting from the Y654S MIDEAS variant. However, we did not observe a general dysregulation of the TGFβ pathway in fibroblasts from Proband 1. Furthermore, downregulation of MAP2K6 might be expected in the patient cells, however, since this gene is not expressed in dermal fibroblasts, such an effect could not be observed. The absence of MAP2K6 expression suggests that dermal fibroblasts may be a limited model system to evaluate the downstream effects of MiDAC variants. Future studies are needed to understand the mechanisms by which MIDEAS variants cause the specific phenotypes described here, and to what extend the TGFβ pathway is involved.

HDACs 1&2 are largely interchangeable paralogues that form the catalytic subunit of six histone deacetylase complexes (MiDAC, SIN3, NuRD, CoREST, MIER and RERE). There appears to be little or no redundancy between these complexes and all are essential for life. Their different functions have been attributed to their different subunit composition, controlling their preferred substrates and their specific recruitment to different genome loci. More recently, an additional common theme appears to be emerging: that the access to the active site of the deacetylase is controlled by subunit components. Recent structures of the human SIN3B complex, the fission yeast SIN3S complex and the budding yeast Rpd3L found that the active site was occupied by a residue from the corepressor protein. In human SIN3B a glutamic acid (E436) is positioned in the active site[32]. Interestingly, a double mutant of Sin3B (E436A/D437A) is no longer able to deacetylate H3K27ac from a modified nucleosome and therefore the loop

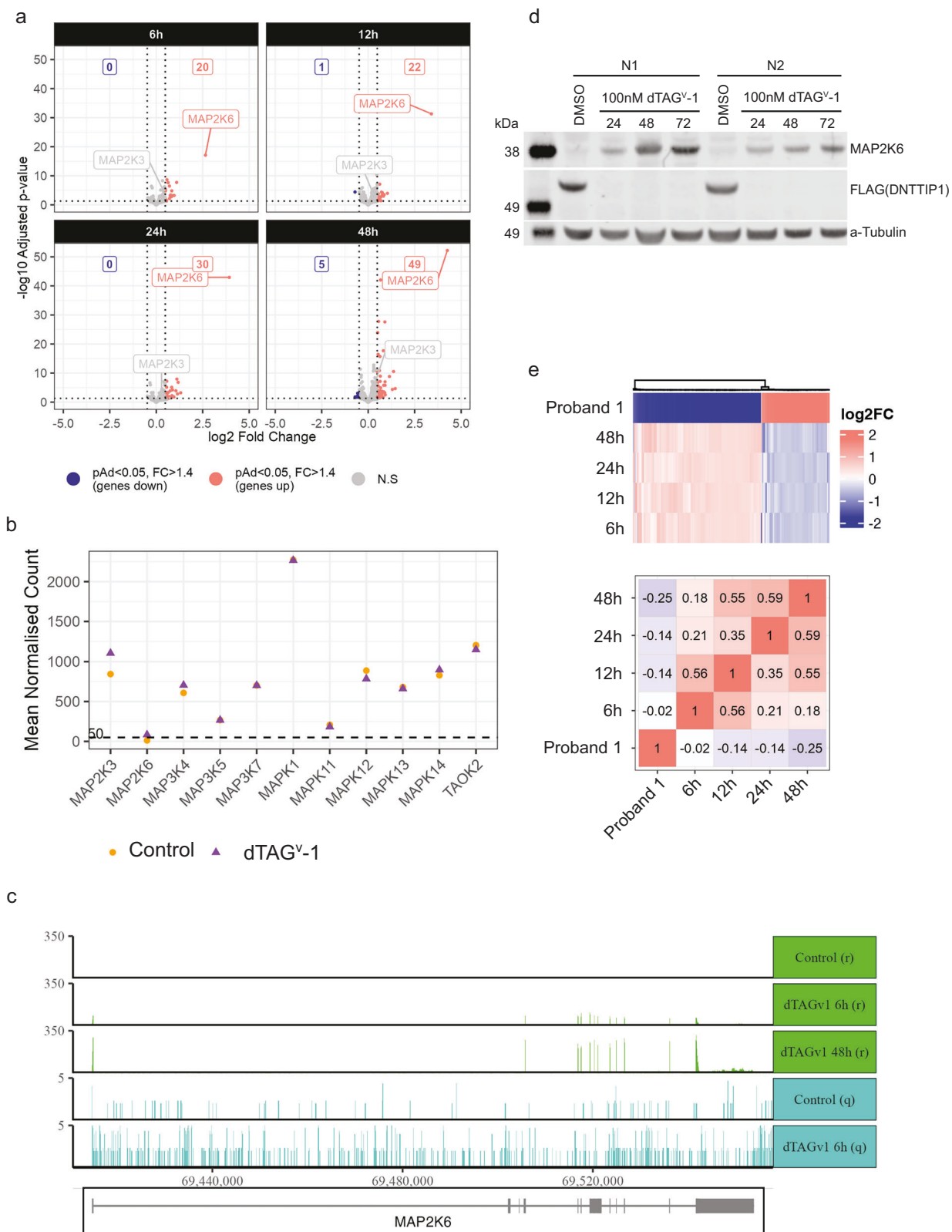

**Fig. 5 | Rapid knockdown of DNTTIP1 in HCT116 cells reveals strong regulation of the MAP2K pathway. a** Volcano plots showing number of differentially expressed genes at 6 h, 12 h, 24 and 48 h following DNTTIP1 knockdown with dTAG$^V$-1. Differential expression was calculated using the Wald test with Benjamini-Hochberg correction. Significant genes were defined as having a p-adjusted value of <0.05 and a fold change of >= 1.2. **b** Average gene counts of MAP2K pathway members in control and 6 h dTAG$^V$-1 treated datasets. **c** Gene locus of

MAP2K6 showing counts from qPROseq (q) and RNAseq (r) experiments with 2/6 h control and dTAG$^V$-1 datasets. **d** Western blot depicting increase in MAP2K6 protein over 72 h following dTAG$^V$-1 treatment (n = 2). The blot was repeated twice with similar results. **e** Heatmap comparing Proband 1 genes (zScore > 2) also present in the HCT116 dataset (115 genes) (log2 fold change limits of −2 and 2 to aid visualisation) and fold change correlation of Proband 1 genes (zScore > 2) also found in HCT116 datasets (115 genes). Source data are provided as a Source Data file.

appears to contribute to substrate recognition. In the *S. pombe* SIN3S HDAC complex, a non-polar phenylalanine side chain (F381 of Pst2) occupies the active site[33] and in the S. cerevisiae Rpd3L HDAC complex there is a leucine side chain (L80 of Rxt2) in one of the Rpd3 active sites[34]. It is not clear whether these amino-acids have a modulatory or auto-inhibitory activity. For the MiDAC complex, we observe that deletion or mutation of the loops has a substantial effect on enzyme activity. For other class I HDAC complexes it remains to be seen if regulatory loops may play a role in substrate specificity and/or enzyme activity.

In conclusion, we present a previously undescribed disorder caused by a p.Tyr654Ser variant in MIDEAS, leading to a hyperactive MiDAC complex. The clinical hallmarks of the disorder are neurodevelopmental delay with speech retardation, joint contractures and thickening of the skin, dysmorphic features with limited facial expression and gastrointestinal problems with early-onset diarrhoea and dysmotility of the gut. The molecular underpinnings of the disorder appear to result from the displacement of an auto-inhibitory or regulatory loop in MIDEAS that covers the HDAC active site. We presume that the loop contributes to substrate specificity either by restricting access to the active site or by being displaced by specific substrates.

## Methods

### Patient consent and genetic analysis
The two families gave written informed consent for diagnostic genetic testing and agreed to the use of data for research purposes in accordance with the Declaration of Helsinki. In the UMC Utrecht, an independent quality check has been carried out to ensure compliance with legislation and regulations (regarding Informed Consent procedure, data management, privacy aspects and legal aspects) (24U-1499).

Clinical whole exome sequence in DNA extracted from whole blood was performed in two different centres for both affected probands and their parents. The two probands, harbouring the same missense variant, were matched using GeneMatcher (https://genematcher.org/). For each affected proband detailed clinical information was obtained through review of the medical records. Written informed consent was obtained from both participants by their parents, and, where applicable, from the affected individual himself. In addition, they provided informed consent for publication of the images in Fig. 1.

### Transcriptomic analysis for Proband 1 fibroblasts
Total RNA was isolated from cultured fibroblasts, grown from a skin biopsy from Proband 1, using NucleoSpin RNA kit (Machery-Nagel). cDNA was generated from fragmented mRNA after PolyA pull down, all using a NEBNext Ultra kit (E7760, New England Biolabs). Resulting cDNA libraries were sequenced using an Illumina NovaSeq X plus platform by the Genome Diagnostics Laboratory at the UMC Utrecht. Sequencing reads were processed using a standardised pipeline.

30 samples were used as a reference set, all dermal fibroblasts derived from unsolved diagnostic cases suspected of a wide variety of monogenic disorders, for which transcriptomic analysis was part of efforts to solve the diagnostic odysseys.

OUTRIDER[35] was used to determine deviant expression, data was analysed and visualised using Voila[36].

### Cell culture
Human HCT116 colon carcinoma cells were cultured in Dulbecco's modified eagle's media (DMEM) high glucose with L-glutamine supplemented with 10 % foetal bovine serum and 1 % PSG (penicillin, streptomycin, glutamine). Cells were detached from plates using TrypLE (Invitrogen). CRISPR/Cas9-engineered HCT116 lines were cultured in the same way as the parental line.

### Generation of DNTTIP1 CRISPR cell line
The endogenous alleles for DNTTIP1 in HCT116 was targeted the using CRISPR/Cas9 system. The targeting gRNA (caccGAAGCACCTCCACAGACCTG; IDT) to exon 13 of DNTTIP1 (ENSG00000101457) was ligated into a Cas9-containing pX330 vector (Addgene). Donor plasmids were generated by ligating homology arms, TEV, FKBP$^{F36V}$, 3xFLAG, P2A and Hygromycin or Neomycin resistance marker into a TOPO vector (Thermo Fisher). HCT116 cells were transfected for 48 h in 6-well plates with 2 μg total DNA of all three plasmids (transfection) or only one of the donor plasmids (negative control) using JetPrime transfection kit (Polyplus) before being transferred to 10 cm dishes. Positive cells were selected using Hygromycin B (2 μl/ml) (Thermo Fisher) and Geneticin (16 μl/ml). Cells were allowed to grow for 14 days before single colonies were transferred into individual wells of a 96-well plate for expansion and genotyping.

### Genotyping
Genomic DNA (gDNA) was extracted from HCT116 cells and CRISPR colonies using a GeneJet kit (Thermo Scientific). The DNTTIP1 exon 13 region was amplified from the gDNA using two primer pairs, targeting 100 bp ('external') and 50 bp ('out') outside of the sequence, analogous to the donor plasmid homology arms. Amplification was carried out in a two-step (re-amplification) PCR reaction using KOD polymerase kit (Sigma-Aldrich). 'External' reaction 1 (F: 5'-GAGAAGGCTATAAATAAGTCACCAGTCC-3'; R: 5'-GGCTCTATACTGTTATGGGTGAGGG-3') produced WT: 893 bp and tagged: 2417/2225 bp (Hygromycin/Neomycin) band sizes. 'Out' reaction 2 (F: 5'-CCCATCCATGAAGCATCTACTGTAG-3'; R: 5'-CGTCCTTTTTTCCTACACCCACTT-3'), using reaction 1 product, resulted in WT: 827 bp and tagged: 2351/2159 bp band sizes.

### RNA sequencing
Cells were seeded into 6-well plates dependent on dTAG$^{v-1}$ treatment duration so there were -1.2 × 10$^6$ cells at time of harvesting. Cells were allowed to settle for 24 h before treatment with 100 nM dTAG$^V$-1 for the indicated time. To harvest, cells were washed with PBS, detached using TrypLE (Invitrogen), diluted in PBS and transferred to nuclease-free 1.5 ml microfuge tubes. Cells were collected by centrifugation at 400 g for 5 min at room temperature, the supernatant decanted and the pellets snap-frozen in liquid nitrogen before storage at -80 °C. Total RNA was isolated by robot and quality checked using the Genomics facility at the University of Leicester. Each experiment was carried out in triplicate.

RNA sequencing was performed by Novogene using a NovaSeq 6000 S4 or NovaSeq Xplus platform using 150 base-pair pair-end sequencing to a depth of 20 million reads. The resulting sequencing files were quality checked using fastqc (v0.12.1)[37]. Sequencing files were aligned using hisat2 (v12.3.0)[38] with standard options to the Ensembl human genome[39] build GRCh38 release 111. Resulting files were converted to binary alignment map (BAM) format, sorted by genomic coordinate and indexed using samtools (v1.17)[40]. Count files were made using LiBiNorm (v2.5)[41] and the GRCh38 release 111 gene transfer format file.

Unless stated otherwise, analysis was performed using R (v4.4.1). DESeq2 (v1.44.0)[42] was used to perform differential expression (Wald test with local fit type and the Benjamini-Hochberg method for P-value correction). The package apeglm (v1.26.1)[43] was used to provide shrinkage estimates for log fold changes. Genes with counts less than 50 in 3 or more samples were excluded. Pathway enrichment was tested using the R package topGO[44],

The previous mouse embryonic fibroblast RNA sequencing data for MIDEAS and DNTTIP1 e13.5 knockout[5] was re-analysed as above, but using Ensembl[39] GRCm39 release 112[39] for alignment and GRCm39 release 112 gene transfer format file for count file formation.

## Quick precision run-on and sequencing (qPROseq) sample preparation

qPROseq was carried out using a published protocol[45]. Cells were treated for 2 or 6 h with 100 nM dTAG$^V$-1 or DMSO, washed twice with ice cold PBS and scraped into ice-cold permeabilization buffer (10 mM Tris/Cl pH 8, 10 mM KCl, 250 mM sucrose, 5 mM MgCl$_2$, 1 mM EGTA, 0.1% IGEPAL CA-630, 0.5 mM DTT, 0.05% Tween-20, 10% glycerol, PIC and SUPERase-In RNase inhibitor (Invitrogen)). Cells were transferred to a falcon and incubated on ice for 5 min. Permeabilised cells were collected by centrifugation at 700 g for 4 min at 4 °C, washed twice with permeabilization buffer without IGEPAL CA-630 and Tween-20 and resuspended in freeze buffer (50 mM Tris/Cl pH 8, 40 % glycerol, 5 mM MgCl$_2$, 1.1 mM EDTA, 0.5 mM DTT, SUPERase-In RNase inhibitor). Cells were counted, collected by centrifugation at 1000 g for 5 min at 4 °C and resuspended in freeze buffer at ~20 × 10$^6$ cells/ml for aliquoting and snap freezing. The run-on reaction was performed by mixing 50 μl permeabilised cells 1:1 with 2 x run-on master mix (10 mM Tris/Cl pH 8, 5 mM MgCl2, 1 mM DTT, 300 mM KCl, 40 μM each of Biotin-11-(CTP/UTP/ATP/GTP), 1% sarkosyl, SUPERase-In RNase inhibitor) and incubating at 37 °C, 750 rpm for exactly 5 min and stopped by the addition of RL buffer. RNA was extracted using a RNA extraction kit (Norgen) and 1 M NaOH used to hydrolyse bases. RNA was precipitated using ethanol and resuspended in nuclease-free water. Following this, reactions for 3' RNA adaptor ligation, streptavidin bead binding, on-bead 5' hydroxy repair, on-bead 5' decapping, on-bead 5' RNA adaptor ligation, TRIzol elution of RNA and off-bead reverse transcription were performed. Test amplification was performed to calculate the required number of cycles for full-scale amplification. DNA was purified after full-scale amplification using AMPure beads, quantified using the qubit HS dsDNA assay and run on a bioanalyzer. All buffers were made in RNase-free water and RNase-free tubes. Reactions were carried out in LoBind RNase-free micofuge tubes and wide-bore, filtered, nuclease-free pipette tips were used for all pipetting.

## qPROseq data processing and analysis

qPRO sequencing was performed by Novogene using a NovaSeq 6000 S4 using 150 base-pair pair-end sequencing to a minimum depth of 50 million reads. Sequencing files were quality checked using fastqc (v0.12.1). Fastp (v0.23.2)[46] was used for preprocessing and removing illumina adaptor sequences before reads aligning to ribosomal RNA were removed using bowtie2 (v12.3.0)[47]. Bowtie2 was used to align to the human hg38 genome. The resulting alignment was filtered to remove reads with a mapq score of less than 10 and converted to BAM format using samtools. UMI-tools (v1.0.1)[48] was used to deduplicate BAM files before samtools was used to sort and index. Deeptools (v3.5.1)[49] bamCoverage was then used to create bigwig files containing forward and reverse reads using a binsize of 1.

The resultant bigwig files were imported into R and count matrices created covering gene bodies using BRGenomics[50] and differential expression performed in the same way as for RNAseq described above.

## Additional R programmes

Data was visualised using the following R programmes: ggplot2 (v3.5.1)[51], ggh4x (v0.2.8)[52], ggcorrplot (0.1.4.1)[53], ComplexHeatmap (v2.20.0)[54], ggcoverage (1.4.0)[55].

## Western blotting

Protein samples mixed with loading buffer and, where applicable, equal amounts loaded. Proteins were separated using NuPAGE 4–12% Bis-Tris gels (Invitrogen) and transferred to nitrocellulose membranes using a semi-dry transfer. Membranes were air-dried at room temperature for 30 min before blocking for 1 h in TBS (20 mM Tris/Cl pH 7.6, 150 mM NaCl) with 0.5% Tween-20 (TBS/T) and 5% milk. Membranes were incubated with primary antibody (msFLAG (1:1000; Sigma); msMAP2K6 (1:1000; Invitrogen); rbTubulin (1:1000; Abcam); msTBP (1:300; SantaCruz)) overnight at 4 °C and for 1 h at room temperature with secondary antibodies (IRdye 680RD goat anti-mouse or IRdye 800CW goat anti-rabbit/ (1:20000); Licor), both diluted blocking buffer. Membranes were imaged using an Odyssey CLx imager and processed using image studio (v5.5.4).

## Protein expression in HEK293F cells

The proteins of the MiDAC complex, DNTTIP1, MIDEAS and HDAC1, were expressed by transient co-transfection using the pcDNA3 vector in HEK293F suspension mammalian cells (Invitrogen). For purification of the complex, WT and mutant MIDEAS constructs were expressed with an N-terminal 10xHis- 3xFLAG tag and a TEV protease cleavage site[12]. The oligonucleotides used to make the MIDEAS mutations are: Y654S (STP) - Forward GCTACCTCCCTCCACGCCGCCCC - Reverse GGGGCGGCGTGGAGGGAGGTAGC; Y654S T655A (SAP) - Forward TTTGAGCTACCTCCCTCCGCGCCGCCCCCCATCCTCAGC - Reverse GCTGAGGATGGGGGGCGGCGCGGAGGGAGGTAGCTCAAA; Y654S T655D (SDP) - Forward TTTGAGCTACCTCCCTCCGATCCGCCCCC-CATCCTCAGC - Reverse GCTGAGGATGGGGGGCGGATCGGAGG-GAGGTAGCTCAAA. Transient transfections were used to express the MiDAC complex, which involved mixing 0.1 mg of each plasmid (0.3 mg DNA total) with 30 ml of PBS (Sigma) and then adding 0.6 ml of 1 mg/ml PEI (Sigma). The suspension was vortexed briefly, incubated for 20 min at room temperature, then added to 300 ml of cells at a density of 1 × 10$^6$ cells/ml. The transfected cultures were harvested 48 h after transfection. For the experiment with flavopiridol, 100 nM flavopiridol was added 48 h after transfection and 2 h before harvesting the cells.

## Purification of the HDAC1/MIDEAS/DNTTIP1 complex

HDAC1/ MIDEAS/DNTTIP1 complexes were purified from 1.2 l of HEK293F cells. After sonication in a buffer containing 50 mM Tris/Cl pH 7.5, 100 mM potassium acetate, 10% (v/v) glycerol, 0.5% (v/v) Triton X-100 and Complete EDTA-free protease inhibitor (Roche) (buffer A), the insoluble material was removed by centrifugation. The complex was then bound to anti-FLAG resin (Sigma), washed twice with buffer A; three times with buffer B (50 mM Tris/Cl pH 7.5, 50 mM potassium acetate, 5% (v/v) glycerol, 0.5 mM TCEP); incubated with 0.5 mg RNaseA for 1 h at 4 °C and then washed five times with buffer B. TEV protease was then used to release the MiDAC complex from the resin overnight on a roller at 4 °C. The complex was gel filtrated on a Superdex-200 column (Cytiva) in 25 mM HEPES pH 7.5, 50 mM potassium chloride, 0.5 mM TCEP.

## Purification of dinucleosome

To make dinucleosome 50 ml HEK293F cells (2 × 10$^6$ cells/ml) were pelleted and resuspended in 25 ml 50 mM Tris/Cl pH 7.5, 4 mM MgCl$_2$, 1 mM CaCl$_2$, 0.5% triton X-100, 0.34 M sucrose, Complete EDTA-free protease inhibitor (Roche) (lysis buffer). The cells were then lysed with a dounce homogeniser. Released nuclei were then centrifuged at 1500 x g, then resuspended in 25 ml cell lysis buffer, homogenised and then centrifuged at 1500 x g. This was repeated three times to wash the nuclei. Washed nuclei were then resuspended in 2 ml cell lysis buffer. Chromatin was digested with 1.5 μl Micrococcal Nuclease (3000 gel units) (New England Biolabs) at 37 °C for 20 min. The digestion was stopped with EDTA to a final concentration of 5 mM and the digested nuclei incubated on ice for 10 min. Nuclei were then centrifuged at 3200 x g then the pellet was resuspended in nuclei lysis buffer (0.2 mM EDTA, 1 mM Tris/Cl pH 7.5, Complete EDTA-free protease inhibitor (Roche)). This was then then centrifuged at 3200 x g and the supernatant concentrated to 0.5 ml, filtered with a 0.22 μm spin filter (Millipore) and gel filtrated on a Superose-S6 (Cytiva) in 25 mM Tris/Cl pH 7.5, 50 mm NaCl, 1 mM EDTA. Fractions were analysed in a 0.7% agarose, 45 mM Tris/Borate gel. A fraction corresponding to the size of a

dinucleosome was concentrated and diluted three times in binding buffer (15 mM Hepes pH 7.5, 50 mM KCl, 5% glycerol, 0.1% triton X-100, 0.5 mM TCEP).

The dinucleosome was then mixed in an equimolar amount with the HDAC1/MIDEAS/DNTTIP1 complex. Glutaraldehyde and formaldehyde were then added to 0.0625% and 0.1% respectively. The complex was then diluted 1/10 or 1/50 with 15 mm Hepes pH 7.5, 50 mM KCl and assessed by negative stain electron microscopy.

## Cryo-EM grid preparation

For cryo-EM dinucleosome and the HDAC1/MIDEAS/DNTTIP1 complex were mixed in an equimolar amount incubated on ice for 1 h then crosslinked with glutaraldehyde (0.0625%) and formaldehyde (0.1%) for 20 min. The complex was then concentrated and diluted three times with 10 mM Hepes pH 7.5, 25 mM KCl, 1 µM inositol hexaphosphate in a 0.5 ml Amicon Ultra centrifugal filter with a 10 kDa MWt cutoff. Finally the dinucleosome-HDAC1/MIDEAS/DNTTIP1 complex was concentrated to 4.55 A260/ml. Then 3 µl sample was applied to a 1.2/1.3 Quantifoil Au300 grid. The grid was blotted for 3 s, before plunge freezing, using a ThermoFisher Vitrobot MKIV at 4 °C and 100% relative humidity, and storing in liquid nitrogen.

## Cryo-EM data collection and processing

Cryo-EM data collection was performed using FEI Titan Krios transmission electron microscope operating at 300 kV using EPU (Thermo Fisher Scientific) (Midlands Regional Cryo-EM Facility in Leicester, UK) and equipped with BioQuantum K3 direct electron detector camera. Images were collected at nominal magnification of 105,000 x in counting mode with a calibrated pixel size of 0.835 A, 2 s exposure time and an accumulative dose of 46.7 e-/A$^2$ and 50 frames per movie. The applied defocus range varied between 1.2 and 2.4 in 0.3 µm intervals.

CryoSparc v4.4.1 was used to process the data (Supplementary Figs. 3 and 4). Particles were autopicked after creating references from particles picked using the blob picker. 2D class averaging was used to obtain an initial cleaned particle set. After heterogeneous classification a clear map for a MiDAC dimer was obtained. Particles from this particle set were then classified in a further round of heterogeneous refinement and the largest class taken forward for 3D refinement to obtain a map to 2.9 Å. Phenix was used to combine focussed maps of the top, middle and bottom of the dimer[56]. A model was built into the combined focussed maps using Modelangelo[57]. This was rebuilt using Coot[58] and refined using Phenix[56].

## Mass spectrometry

Gel bands were incubated in 80 µl 50 mM ammonium bicarbonate for 5 min at RT. The ammonium bicarbonate was aspirated, and 80 µl acetonitrile added. The acetonitrile was aspirated after 5 min. The process was repeated until no further stain was removed. After the final organic wash the gel pieces were dehydrated and trypsin (Promega) (0.16 µg in 15 µl 50 mM ammonium bicarbonate) was added to each sample and incubated at 37 °C overnight. Peptides were extracted using 80 µl 90% TFA (0.2%), 10% acetonitrile for 1 h at RT, then the solutions are transferred into low-binding Eppendorf tubes and concentrated to dryness in a vacuum concentrator at 70 °C. The samples were then dissolved in 20 µl of injection solvent (95% formic acid (0.2%), 5% acetonitrile). 3 µl was injected.

Tryptic peptides were separated on an Ultimate 3000 RSLC Nano HPLC system (Dionex/Thermo Fisher Scientific, Bremen, Germany). Samples were loaded onto a cartridge-based trap column, using a 300 µm x 5 mm C18 PepMap (5 µm, 100 Å) and then separated using Easy-Spray pepMap C18 column (75 µm x 50 cm); with a gradient from 3 to 10% B for 10 min, 1–50% B for 37 min, 50–90% for 9 min and 90–3% for 26 min, where mobile phase A was 0.1% formic acid in water and mobile phase B, 80% formic acid (0.1%), 20% acetonitrile. Flow rate was

0.3 µl/min. The column was operated at a constant temperature of 40 °C. The NanoHPLC system was coupled to a Q-Exactive mass spectrometer (ThermoScientific, Bremen, Germany). The Q-Exactive was operated in the data-dependent top10 mode; full MS scans were acquired at a resolution of 70,000 at m/z 200 to 2000, with an ACG (ion target value) target of 1e6, maximum fill time of 50 ms. MS2 scans were acquired at a resolution of 17,500 with an ACG target of 1e5, maximum fill time of 100 ms. The dynamic exclusion was set at 30.0 s, to prevent repeat sequencing of peptides. The data was analysed using PEAKS Studio X (BSi, Canada) then compiled and visualised using Scaffold (Proteome Software, Oregon).

## HDAC assays using NMR

NMR HDAC assays were performed on 600 µl samples (in standard 5 mm NMR tubes) of 100 µM H3K9ac peptide (ARTKQTAR[K-Ac]STGGKAPRKQLA) (purchased from GenScript) with 50 nM HDAC1 for each complex in 10 mM Tris/Cl pH 7.5, 50 mM KCl, 0.5 mM TCEP, 100 µM inositol hexaphosphate and 5% D$_2$O. All spectra were recorded on a 600 MHz Bruker AVIII spectrometer using a 5 mm TCI cryoprobe operating at 293 K. The $^1$H spectra were collected as a pseudo 2D experiment with a 1 sec acquisition time and a 1.5 s relaxation delay between each transient, with each timepoint taking 20 sec to acquire. Typical acquisition times of the pseudo 2D experiments was 1 hr. Each experiment was started approximately 95 sec post addition of the complex with each subsequent time point at 10 s intervals. All NMR data was processed and analysed using Topspin version 3.6.5. Initial rates of reaction were calculated between 95 and 995 s using Prism (GraphPad).

## Reporting summary

Further information on research design is available in the Nature Portfolio Reporting Summary linked to this article.

## Data availability

Patient exome sequencing and fibroblast transcriptomic data are not publicly available due to privacy/ethical restrictions. Raw and processed RNA sequencing and nascent sequencing files have been submitted to GEO under the following accession codes: GSE297959; GSE297958. The MIDEAS and DNTTIP1 MEF datasets were reanalysed from GSE144748. The coordinates for the MiDAC model are available from the PDB under the accession code 9R4I. The EM maps are available from EMDB under the following accession codes: consensus map EMD-53563; focussed maps EMD-53564, EMD-53565, EMD-53566; combined focussed map EMD-53567. The mass spectrometry proteomics data have been deposited to the ProteomeXchange Consortium via the PRIDE partner repository with the dataset identifier PXD067661. Source data are provided with this paper.

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

## Acknowledgements

We thank the Probands and their families. We are grateful to the Midlands Regional CryoEM facility, hosted by the Institute for Structural and Chemical Biology at the University of Leicester, including support from Dr Emma Hesketh, Dr Christos Savva, Dr TJ Ragan and Dr Claudia Lancey. Expression plasmids were prepared by the PROTEX facility at the University of Leicester with support from Dr Xioawen Yang and Sharon Munday. We are grateful to the Mass Spectroscopy and NMR facilities hosted by the Institute for Structural and Chemical Biology at the University of Leicester. We thank H van Deutekom and M Nelen, Dept. of Genetics, UMC Utrecht for support and research facilities. JWRS is funded by a Wellcome Trust Investigator grant (222493/Z/21/Z).

## Author contributions

L.F. performed the biochemistry, structural biology & enzymology. K.S. generated the HCT116 cell lines for complex degradation and performed multiple omics analyses and biochemistry. R.E.T. performed comparative analyses of the patient RNAseq data with fibroblasts from the MIDEAS KO mice and with the HCT116 cells following DNTTIP1 degradation. S.K. designed the research protocol, consent forms, participated in recruitment and phenotyping of the probands. O.G. designed and guided the CRISPR experiments to tag DNTTIP1. F.W.M. performed the NMR experiments to measure HDAC activity of various MiDAC complexes. L.B., E.G., and K.G. interpreted the results of next generation sequencing. R.J.J. performed the mass-spectrometry to determine PTMs of the various MIDEAS proteins. P.B. provided expertise and support for the qPROseq experiments. D.J. was involved in the inclusion and phenotyping of the probands. P.T. was involved in the design of the study, supervised the design of the research protocol, inclusion and phenotyping of the probands. P.M.H. initiated the collaboration between clinicians and structural biologists and identified the "gene-deserts" correlated with key functional areas. R.H.J. supervised and collected clinical RNAseq data and interpreted the results. J.W.R.S. supervised the structural biology, biochemistry, omics of the MEF KO and CRISPR degradation experiments. All authors contributed to preparing and reviewing the figures and manuscript.

## Competing interests

The authors declare no competing interests.
