## [Transparent Peer Review file · Nature Communications]

A de novo missense variant in MIDEAS results in increased deacetylase activity of the MiDAC HDAC complex causing a neurodevelopmental syndrome.

Corresponding Author: Professor John Schwabe

Version 0:

Reviewer comments:

Reviewer #1

(Remarks to the Author)

The present manuscript reports the human neurodevelopmental disorder caused by a de novo heterozygous variant (Y654S) in the Mideas gene. The authors describe the phenotypes linked to this mutation. The authors determine the cryo-EM structure of the MiDAC dimer and demonstrate that Y654 resides in a conserved auto-inhibitory loop that covers the HDAC1 active site and modulates the deacetylase activity. The authors use NMR based HDAC assays to confirm the increase in deacetylase activity for the Y654S mutant. The mutation creates a CDK phosphorylation site leading to phosphorylation of Y654S or T655. Further, transcriptomic analysis of patient-derived fibroblasts reveals alteration in gene expression. However, the study requires major improvements in chromatin localization of MiDAC complex, gene expression analysis, and mechanistic validation to strengthen its conclusions.

Major concerns

1. The authors have previously reported the structure of the MiDAC complex and now present a more detailed structure in this study. A detailed comparison between the current and prior structures is necessary.
2. The authors report phosphorylation of Y654S or T655 in the mutant MiDAC complex (Figure 4e) and propose that CDKs are responsible. To substantiate this claim, the authors should perform inhibition of CDKs using drugs or genetic approaches to confirm CDK involvement in phosphorylating the mutant Mideas.
3. The authors propose that the Y654S mutation results in hyperactive MiDAC complex impacting gene expression. However, the link between MiDAC complex and its chromatin function needs to be strengthened. Techniques such as CHIPseq/CUT&RUN/CUT&TAG could be used to demonstrate the chromatin localization of the MiDAC complex (or at least a component of the complex) and the effect on histone acetylation/deacetylation (H3K27ac). These experiments could be conducted in a cell culture model (HEK293/HCT116) using lentiviral/retroviral/genomic integration to study the gain of function effects of Y654S mutation.
4. The authors claim a reciprocal relationship between expression profiles in patient fibroblasts and engineered loss-of-function models, suggesting that Y654S causes MiDAC hyperactivity. This claim requires clearer presentation of the affected genes in both systems. A detailed list of differentially expressed genes, along with their functional roles and GO terms, should be provided to support the proposed etiology.
5. Figure 5e lacks sufficient explanation. The authors should clarify the blue and red bars, specify the number of genes represented, and list those showing reciprocal correlation.
6. The use of a MiDAC degradation approach in HCT116 cells is unclear, as it does not directly elucidate the mechanism of the Y654S mutation. The authors identify MAPK2K6 as MiDAC regulated gene, but should further demonstrate whether MiDAC physically localizes to MAP2K6 regulatory elements.
7. Figure 5d - Quality of western blot needs to be improved.

Reviewer #2

(Remarks to the Author)

The work from Fairall and colleagues describes a newly identified case of a multisystem disorder characterized by delayed speech development, joint contractures, dysmorphic features and dysmotility of the gut. Using clinical trio-exome sequencing the authors identified in the MIDEAS 110 c.1961A>C (p.Tyr654Ser) mutation the one putatively responsible for

the phenotype. The MIDEAS protein is a core component of the MiDAC histone deacetylase complex that is additionally constituted by HDAC1 and DNMT1 proteins. The authors also managed to solve a cryoEM structure of the MiDAC complex that is different from the one previously published by the same authors because it contains longer portions of the MIDEAS proteins (aa 653-670 and 631-637). The latter regions include the T654 that is mutated in the identified syndrome and have been described in this study to be key for regulating the activity of the MiDAC complex. So, the structure has great novelty. The authors also showed nice activity assays on mutants and phospho-mimetic mutants that well complement the structural observations. Lastly, using a PROTAC based strategy the authors showed that depletion of the MIDAS protein leads to upregulation of MAP2K6. Overall, this is a very nice and interesting story that combines patients, genomics approaches, structural and molecular biology. The text is written well and can be easily followed. Before publication, it nevertheless requires some major and minor points to be addressed. My overall main criticism is about the interpretability of the cryoEM related figures, that to me are not conveying the important mechanistic insight that have been found and that are well described in the text. In more detail:

- Figure 3A: as it is, it's not really clear. would have preferred to see a cartoon representation of the protein, close to the map (non transparent map, maybe colored the same way as the cartoon).
- Supplementary figure 3A: as it is, is not really explaining the workflow used to obtain the high resolution map and should be better integrated and described.
- In supplementary figure 3A, the FSC shows a little bump in the "corrected" curve. Do the author see or expect flexibility of the molecule? Is it due to focused map? Also, the orientation distribution suggests some preferential orientation of the MiDAC complex. Is this affecting the map quality? Would be useful to have very short comment on these in the main text.
- If I well understood, the purpose of the authors was to solve the structure of the MiDEAS complex bound to nucleosome, but in the end the dataset resulted in two separate maps (unfortunately). I think this is essential to be mentioned at the beginning of the cryoEM description (line 173).
- The cryoEM sample has been crosslinked but the processing resulted in two separate maps: MiDAC and nucleosomes. Why? Also, considering that all the sample prep procedure was tailored on dinucleosome, I expected to see a dinucleosome map, instead there is a mononucleosome, why?
- Figure 3B and 3C are a bit messy. Please avoid using red circles because color blind might not see them. In fig 3B, instead of a circle please use a label to indicate the active site.
- Most of the residues described from line 174 to 179 are not shown in the structural figures. Would be helpful to have them.
- Considering that the DNMT1 regions just mentioned above are loops that structures upon binding with the other MIDEAS component, it would be helpful to have a close-up view (maybe in supplementary) showing the EM densities in that portion.
- Figure 4A: would be nice to have aminoacid numbering above the sequence as a reference
- Apologize for keep repeating, but the figure 4B is not clear. Would be beneficial to show the loops in cartoon and only the key residues in sticks. the interaction well described in the text between I659 from MiDEAS the two phenylalanines in HDAC1 (F150 and F205) is not well represented in main figure, but they are in the supplementary figure 5c-f. I would consider to add the suppl fig 5d without the electron density to the main text. Great to see EM densities on these regions in suppl 5.
- Line 283: from my understanding of the work, MAP2K3 is upregulated only at 2h (as shown in suppl fig 7c) and has not been validated further. Please modify the sentence specifying this or validate further via WB or remove it.

Minor:

Line 66 not clear sentence

Supplementary figure 2: "before" better if called "input"

Line 172: better to add "compared to the previously determined one"

Link in text to suppl fig 4 is missing.

Reviewer #3

(Remarks to the Author)

The manuscript by Fairall et al. starts from a clinical observation -two patients having a complex syndrome with overlapping phenotypes show the same mutation in the MiDAC complex- and examines at structural and mechanistic level how this mutation may explain the phenotypes. The overall logic of the manuscript is well-constructed. However, several aspects require further clarification and improvement.

Major Concerns:

1) Gene expression changes in patient samples vs MIDAS KO cells

The gene expression data should be presented as heatmaps and GSEA, in addition to the pathway analysis presented.

2) Cryo-EM structure and auto-inhibitory loop:

The Cryo-EM structure of the MiDAC complex with an extended MIDEAS isoform provides additional structural information, but it does not represent a major advancement in the field. One of the novel findings—the identification of an "auto-inhibitory" loop spanning residues 653–670—is intriguing, but the data presented are insufficient to conclusively support its inhibitory function. In Fig 4, the authors delete the so-called inhibitory loop, which is not an optimal design. Truncated protein may differ a lot from wild type in many aspects, like solubility and stability, and this may also affect the deacetylation activity. A loop mutant (mutating inhibitory loop residues to alanine) needs to be investigated.

Besides, the in vitro assay using a synthetic peptide may introduce bias and does not convincingly reflect physiological regulation of deacetylase activity. To strengthen the claim of an auto-inhibitory function, a cellular assay using genetically

modified cell lines (e.g., rescue experiments in MIDEAS knockout cells with either wild-type or loop-mutated MIDEAS) would provide more robust functional evidence, together with the natural substrate's acetylation, e.g., acetyl-histone H3/H4). Ideally, this would be done at proteome level by performing an acetylome in the different cell lines.

3) Logical flow and interpretation of the final section of "Results":

The final section of the results lacks coherence and introduces a shift in focus toward the rapid degradation of the MiDAC complex, which appears tangential to the manuscript's central theme. The paragraph beginning with "To further investigate the role of the MiDAC complex..." is inconsistent with the preceding sections, which focus on a novel regulatory mechanism rather than the broader function of the complex—an area already covered extensively in prior studies.

Additionally, it is unclear whether the degradation of the MiDAC complex occurs under physiological conditions or only in experimental settings. The sentence on lines 256–261, stating that "...showed a striking reciprocal relationship that becomes stronger with time...", appears to contradict the data in Figure 5e, where the value is negative and decreases over time, suggesting the genes upregulated in the MiDAC-deficient model differ significantly from those seen in patient samples. The authors should address these points carefully.

4) While the cryo-EM and NMR data strongly suggest hyperactivation of Y654S, additional cellular assays would strengthen the claim. In addition, it would be interesting to examine how this hyperactivation reacts to HDAC specific inhibitor, e.g., SAHA or TSA.

5) Depletion of MiDAC by DNTTIP1 degradation is not convincing, as MiDAC has multiple subunits. The result observed may be a biased outcome from DNTTIP1 degradation. To further confirm the observation is really coming from MiDAC malfunction, the same experiments should be performed following depletion of other subunit(s), or through a mutation of MiDAC catalytic activity.

6) The connection between MiDAC hyperactivity and MAP2K6 upregulation is intriguing but needs more validations. Is the p38 phosphorylation altered Y654S-expressing cells? Does p38 inhibition rescues any of the cellular phenotypes observed (e.g., proliferation defects, cytokine secretion)?

Specific points:

- 1) Figure 5c: The labels "control r" and "control q" need clarification.
- 2) Figure 5e: The meaning of the color coding should be explained clearly in the legend.
- 3) Figure 5d: To validate successful fractionation of cytosolic and nuclear components, the western blot should be presented as a single, horizontal blot. This would clearly demonstrate that nuclear markers are absent from the cytosolic fraction and vice versa.
- 4) Figure 4c, the Y and X-axis labels are too small to evaluate.
- 5) Line 259 – Interpretation Issue: The conclusion that "the variant (which variant?) leads to a hyperactive MiDAC" is not well-supported. Which variant should be clearly specified, and the logic connecting the data to this conclusion must be more rigorously explained.
- 6) Line 111: Full name of gnomAD?
- 7) Line 141: is the clinical RNAseq data publicly available or generated by the author? Please specify.
- 8) line 214: Fig 4e was referred to before Fig 4c and d. The organization of the figure should be changed.
- 9) The discussion of auto-inhibitory loops in other HDAC complexes (SIN3B, Rpd3L) is interesting but superficial. The authors could explore more by comparing the MIDEAS loop with known regulatory mechanisms in other HDAC complexes.
- 10) All abbreviations should be defined upon their first appearance in the text.

Reviewer #4

(Remarks to the Author)

Version 1:

Reviewer comments:

Reviewer #1

(Remarks to the Author)

All of my concerns have been satisfactorily addressed by the authors.

Reviewer #2

(Remarks to the Author)

The authors answered to all my points.

Reviewer #3

(Remarks to the Author)

The manuscript by Fairall and colleagues identifies missense variants in the MIDEAS complex which result in increased deacetylase activity of the MIDAC HDAC complex and translates in a neurodevelopmental syndrome.

I appreciate the efforts made by the authors to address various points. While I do not want to diminish the author's work for the revision, I still find that the data pertaining to the function of the auto-inhibitory loop are not sufficiently convincing.

Further, given the small number of samples analyzed, an experiment engineering the loop mutations in a cell line would be very useful to validate the physiology of the findings. Using HCT116 cells would be good (likewise, ChIP data from HCT116 cells would seem to be more useful than data from the literature in mESC cells).

Reviewer #4

(Remarks to the Author)

Response to reviewers comments

REVIEWER COMMENTS

Reviewer #1 (Remarks to the Author):

The present manuscript reports the human neurodevelopmental disorder caused by a de novo heterozygous variant (Y654S) in the Mideas gene. The authors describe the phenotypes linked to this mutation. The authors determine the cryo-EM structure of the MiDAC dimer and demonstrate that Y654 resides in a conserved auto-inhibitory loop that covers the HDAC1 active site and modulates the deacetylase activity. The authors use NMR based HDAC assays to confirm the increase in deacetylase activity for the Y654S mutant. The mutation creates a CDK phosphorylation site leading to phosphorylation of Y654S or T655. Further, transcriptomic analysis of patient-derived fibroblasts reveals alteration in gene expression.

However, the study requires major improvements in chromatin localization of MiDAC complex, gene expression analysis, and mechanistic validation to strengthen its conclusions.

We thank the reviewer for their constructive comments and we have modified the manuscript to take these into account.

Major concerns

1. The authors have previously reported the structure of the MiDAC complex and now present a more detailed structure in this study. A detailed comparison between the current and prior structures is necessary.

We have added supplementary figure 5 to highlight the differences between the current structure and our previous structure.

2. The authors report phosphorylation of Y654S or T655 in the mutant MiDAC complex (Figure 4e) and propose that CDKs are responsible. To substantiate this claim, the authors should perform inhibition of CDKs using drugs or genetic approaches to confirm CDK involvement in phosphorylating the mutant Mideas.

Thank you for this insightful suggestion.

We have treated MiDAC expressing cells with the CDK2/9 inhibitor, flavopiridol hydrochloride @100nM, for 2 hours prior to harvesting cells. Following 2 hours of treatment, we observe a c.22% decrease in phosphorylation of the auto-inhibitory loop (Figure 4c).

3. The authors propose that the Y654S mutation results in hyperactive MiDAC complex impacting gene expression. However, the link between MiDAC complex and its chromatin function needs to be strengthened. Techniques such as ChIPseq/CUT&RUN/CUT&TAG could be used to demonstrate the chromatin localization of the MiDAC complex (or at least a component of the complex) and the effect on histone acetylation/deacetylation (H3K27ac). These experiments could be conducted in a cell culture model (HEK293/HCT116) using lentiviral/retroviral/genomic integration to study the gain of function effects of Y654S mutation.

This is an excellent suggestion.

To address this point, we have now analysed published data that used ChIP to identify the genomic locations of the MiDAC complex Mondal et al (2020) and Wang et al. (2022).

These published data strongly support our findings since the MiDAC complex was observed on both MAP2K6 and MAP2K3 genes (supplementary figure 8). The complex was also observed on genes showing reciprocal regulation between the Patient fibroblasts and our HCT116 cells. Knockout of the MiDAC complex (DNTTIP1) resulted in increased acetylation of H3K27 and H4K20.

Although these data do not directly address the effect of Y654S on the binding and acetylation status at these loci, we suggest that our combined observations (enhanced mutant acetylation activity (Fig. 4e), overall reciprocal regulation in mutant vs knock-out (Fig. 5e)) together with the published CHIP data, support our proposed mechanism.

To directly test the effect of the mutation on chromatin binding and local acetylation status would be highly challenging since it would be difficult to discriminate between the effects of perturbed expression (resulting from the gene editing) and variant-specific effects. Therefore, whilst certainly interesting, we believe that such a challenging experiment is beyond the of the current manuscript.

4. The authors claim a reciprocal relationship between expression profiles in patient fibroblasts and engineered loss-of-function models, suggesting that Y654S causes MiDAC hyperactivity. This claim requires clearer presentation of the affected genes in both systems. A detailed list of differentially expressed genes, along with their functional roles and GO terms, should be provided to support the proposed etiology.

We now include a list of genes that are reciprocally regulated between the Proband 1 and the MEFs derived from the MIDEAS knockout mice (Supplementary figure 1b). We also include in the same file the list of the reciprocally regulated genes between the Proband 1 and the HCT116 cells in which DNTTIP1 has been degraded.

5. Figure 5e lacks sufficient explanation. The authors should clarify the blue and red bars, specify the number of genes represented, and list those showing reciprocal correlation.

Thank you for pointing this out. We have added a scale for the colour code in figure 5e and the number of genes in the figure legend. We also now include, in a supplementary file, the reciprocally regulated genes between the Proband 1 and the HCT116 model.

6. The use of a MiDAC degradation approach in HCT116 cells is unclear, as it does not directly elucidate the mechanism of the Y654S mutation. The authors identify MAPK2K6 as MiDAC regulated gene, but should further demonstrated whether MiDAC physically localizes to MAP2K6 regulatory elements.

We are grateful for pointing out that we had not made clear our rationale for this experiment. We have now adjusted the text to explain that we reasoned that if our hypothesis that the mutant results in a hyperactive MiDAC complex in the Proband 1, then degradation of the complex in a human cell-line should result in opposite changes in gene expression to those observed in the patient. We were pleased that our reasoning was supported by the observed

reciprocal regulation (Y654S activating mutation vs. complex depletion) despite the differences between primary fibroblasts vs. engineered colon cancer cells line.

Regarding the genomic localization of MiDAC, please see point 3.

7. Figure 5d - Quality of western blot needs to be improved.

We have repeated the Western blot with two replicates to confirm degradation of DNTTIP1 in the engineered HCT116 cells and the resulting increase in MAP2K6 protein.

Reviewer #2 (Remarks to the Author):

The work from Fairall and colleagues describes a newly identified case of a multisystem disorder characterized by delayed speech development, joint contractures, dysmorphic features and dysmotility of the gut. Using clinical trio-exome sequencing the authors identified in the MIDEAS 110 c.1961A>C (p.Tyr654Ser) mutation the one putatively responsible for the phenotype. The MIDEAS protein is a core component of the MiDAC histone deacetylase complex that is additionally constituted by HDAC1 and DNTTIP1 proteins. The authors also managed to solve a cryoEM structure of the MiDAC complex that is different from the one previously published by the same authors because it contains longer portions of the MIDEAS proteins (aa 653-670 and 631-637). The latter regions include the T654 that is mutated in the identified syndrome and have been described in this study to be key for regulating the activity of the MiDAC complex. So, the structure has great novelty. The authors also showed nice activity assays on mutants and phospho-mimetic mutants that well complement the structural observations. Lastly, using a PROTAC based strategy the authors showed that depletion of the MIDAS protein leads to upregulation of MAP2K6. Overall, this is a very nice and interesting story that combines patients, genomics approaches, structural and molecular biology. The text is written well and can be easily followed. Before publication, it nevertheless requires some major and minor points to be addressed. My overall main criticism is about the interpretability of the cryoEM related figures, that to me are not conveying the important mechanistic insight that have been found and that are well described in the text.

We are grateful to the reviewer for their thorough evaluation and kind words. We have made substantial adjustments according to their comments.

In more detail:

- Figure 3A: as it is, it's not really clear. would have preferred to see a cartoon representation of the protein, close to the map (non transparent map, maybe colored the same way as the cartoon).

Thank you for this suggestion. We have remade figure 3 accordingly.

- Supplementary figure 3A: as it is, is not really explaining the workflow used to obtain the high resolution map and should be better integrated and described.

We have amended this figure to clarify the steps involved in getting to the high resolution map.

- In supplementary figure 3A, the FSC shows a little bump in the “corrected” curve. Do the author see or expect flexibility of the molecule? Is it due to focused map? Also, the orientation distribution suggests some preferential orientation of the MiDAC complex. Is this affecting the map quality? Would be useful to have very short comment on these in the main text.

The map represents the core of the MiDAC dimer and, at low contour levels, we can see low quality density in the map for the flexible DNA binding domains. Using masks the map was improved to create the composite map. The FSC curve in the supplementary figure was from the non-uniform map before the composite focussed map was made. The blip is seen in all the corrected FSC curves and is due to phase randomisation (<https://discuss.cryosparc.com/t/fsc-curve-during-local-refinement-looks-weird/10006>). We have added a sentence to further clarify that we cannot see the whole complex at high resolution.

- If I well understood, the purpose of the authors was to solve the structure of the MiDEAS complex bound to nucleosome, but in the end the dataset resulted in two separate maps (unfortunately). I think this is essential to be mentioned at the beginning of the cryoEM description (line 173).

We have now explained in the manuscript that due to flexibility of the complex we were unable to see either the DNA-binding domains of DNTTIP1 or the interaction of MiDAC with a nucleosome.

- The cryoEM sample has been crosslinked but the processing resulted in two separate maps: MiDAC and nucleosomes. Why?

We think that the DNA binding domains are too flexibly linked to the core dimer to allow determination of a high resolution map of the complex bound to nucleosome. The best 3D class averages that we were able to obtain are shown below.

Also, considering that all the sample prep procedure was tailored on dinucleosome, I expected to see a dinucleosome map, instead there is a mononucleosome, why?

The linker between adjacent nucleosomes is likely to be highly flexible so that the relationship between nucleosomes would be highly variable. We were hoping that a MiDAC:di-nucleosome complex might adopt a more rigid conformation.

- Figure 3B and 3C are a bit messy. Please avoid using red circles because color blind might not see them. In fig 3B, instead of a circle please use a label to indicate the active site.

We have relabelled these figures according to the suggestion.

- Most of the residues described from line 174 to 179 are not shown in the structural figures. Would be helpful to have them.

These were previously illustrated in supplementary figure 5. We have now moved these figures to main figure 3.

- Considering that the DNTTIP 1 regions just mentioned above are loops that structures upon binding with the other MiDEAS component, it would be helpful to have a close-up view (maybe in supplementary) showing the EM densities in that portion.

We now show this in main figure 3.

- Figure 4A: would be nice to have aminoacid numbering above the sequence as a reference

We have added numbering to this alignment.

- Apologize for keep repeating, but the figure 4B is not clear. Would be beneficial to show the loops in cartoon and only the key residues in sticks. the interaction well described in the text between I659 from MiDEAS the two phenylalanines in HDAC1 (F150 and F205) is not well represented in main figure, but they are in the supplementary figure 5c-f. I would consider to add the suppl fig 5d without the electron density to the main text. Great to see EM densities on these regions in suppl 5.

We have amended figures 3 and 4b accordingly.

- Line 283: from my understanding of the work, MAP2K3 is upregulated only at 2h (as shown in suppl fig 7c) and has not been validated further. Please modify the sentence specifying this or validate further via WB or remove it.

Based on published CHIP data (Mondal et al. (2020) and Wang et al. (2022)) we now show the MiDAC complex localizes to both MAP2K6 and MAP2K3 (supplementary figure 8), strengthening the regulation of MAP2K3 by MiDAC.

Minor:

Line 66 not clear sentence

We have re-worded this.

Supplementary figure 2: “before” better if called “input”

We have re-labelled this “Load” to indicate it is the sample that was loaded onto the gel filtration column.

Line 172: better to add “compared to the previously determined one”

We have made this modification.

Link in text to supp fig 4 is missing.

This is now included.

Reviewer #3 (Remarks to the Author):

The manuscript by Fairall et al. starts from a clinical observation -two patients having a complex syndrome with overlapping phenotypes show the same mutation in the MiDAC complex- and examines at structural and mechanistic level how this mutation may explain the phenotypes. The overall logic of the manuscript is well-constructed. However, several aspects require further clarification and improvement.

We thank the reviewer for their thorough evaluation of our manuscript and suggestions for improvement. We have made substantial modification according to these suggestions.

Major Concerns:

1)Gene expression changes in patient samples vs MIDAS KO cells

The gene expression data should be presented as heatmaps and GSEA, in addition to the pathway analysis presented.

Thank you for these suggestions. However, with data from only one patient (N=1), we feel a GSEA analysis would not provide reliable insights. We have included a heat map in figure 2 and a gene list / volcano plot in supplementary figure 1 to more clearly present our data.

2)Cryo-EM structure and auto-inhibitory loop:

The Cryo-EM structure of the MiDAC complex with an extended MIDEAS isoform provides additional structural information, but it does not represent a major advancement in the field. One of the novel findings—the identification of an “auto-inhibitory” loop spanning residues 653–670—is intriguing, but the data presented are insufficient to conclusively support its inhibitory function. In Fig 4, the authors delete the so-called inhibitory loop, which is not a optimal design. Truncated protein may differ a lot from wild type in many aspects, like solubility and stability, and this may also affect the deacetylation activity. A loop mutant (mutating inhibitory loop residues to alanine) needs to be investigated.

Like the reviewer, we were also concerned that deletion of the loop might affect the solubility / stability of the complex. We were reassured, however, that the expression levels, and purification of the two proteins was essentially identical (Supplementary figure 6). This was true of the three mutants that we created (YTP -> STP, SAP, SDP) (Supplementary figure 6).

Although we indeed use full truncation of the loop as one of the mutants in our analysis, we also tested several amino acid substitutions (SAP, STP, SDP) to investigate the effects of the Y654S variant. As this is the main focus of our study, we are not sure how mutating the loop to alanines would add to our conclusions. Importantly, the STP mutant (equivalent to the Y654S human variant) results in an >3 fold increase in deacetylation rate. We have elaborated on our choice of mutants in the text to clarify our rationale.

Besides, the in vitro assay using a synthetic peptide may introduce bias and does not convincingly reflect physiological regulation of deacetylase activity. To strengthen the claim

of an auto-inhibitory function, a cellular assay using genetically modified cell lines (e.g., rescue experiments in MIDEAS knockout cells with either wild-type or loop-mutated MIDEAS) would provide more robust functional evidence, together with the natural substrate's acetylation, e.g., acetyl-histone H3/H4). Ideally, this would be done at proteome level by performing an acetylome in the different cell lines.

This is an insightful comment. We have added an analysis of published CHIPseq data to demonstrate MiDAC directly localizes to the loci of genes with are differentially expressed both in our depletion models and the patient fibroblasts (Supplementary Figures 8-10). Importantly, these genes show reciprocal effects, supporting an increased activity of the Y654S mutant (Figures 2 and 5). In summary, Y654S shows increased acetylation activity *in vitro*, and patient fibroblasts show reciprocal expression effects of genes where MiDAC localizes to the chromatin, associated with increased local acetylation. With the addition of the CHIPseq data, we therefore addressed the gap in physiological function. Although we agree the proposed experiment is highly interesting, we kindly ask the reviewer for their understanding this elaborate experiment is out of scope of the current manuscript.

3) Logical flow and interpretation of the final section of “Results”:

The final section of the results lacks coherence and introduces a shift in focus toward the rapid degradation of the MiDAC complex, which appears tangential to the manuscript's central theme. The paragraph beginning with “To further investigate the role of the MiDAC complex...” is inconsistent with the preceding sections, which focus on a novel regulatory mechanism rather than the broader function of the complex—an area already covered extensively in prior studies.

Thank you for correctly pointing out the lack of a clear rationale. We have reworded this section to enhance the logical flow making clear that we were seeking to substantiate our hypothesis that the mutation results in a hyperactive complex *in vivo* by demonstrating reciprocal gene expression effects when the complex is degraded in a human cell line.

Additionally, it is unclear whether the degradation of the MiDAC complex occurs under physiological conditions or only in experimental settings. The sentence on lines 256–261, stating that “...showed a striking reciprocal relationship that becomes stronger with time...”, appears to contradict the data in Figure 5e, where the value is negative and decreases over time, suggesting the genes upregulated in the MiDAC-deficient model differ significantly from those seen in patient samples. The authors should address these points carefully.

We generated an engineered cell line to enable *in vivo* depletion of the DNTTIP1 protein through PROTAC induced degradation. We now emphasize this in the text. The negative values indicate negative correlation and thus reciprocal effects. The decreasing values therefore reflect an increased reciprocal effect. To clarify, we adjusted the text accordingly. Of note, if the gene sets would differ between both models, we expect the correlation values to regress to zero (no correlation).

4) While the cryo-EM and NMR data strongly suggest hyperactivation of Y654S, additional cellular assays would strengthen the claim. In addition, it would be interesting to examine how this hyperactivation reacts to HDAC specific inhibitor, e.g., SAHA or TSA.

MiDAC represents less than 5% of the HDAC1 containing complexes in cells (Alshehri et al. bioRxiv 2025, DOI:10.1101/25.02.24.639909). We therefore expect it will be very difficult to

determine the effect of HDAC inhibitors on the mutant activity. We now mention this in the introduction.

5) Depletion of MiDAC by DNTTIP1 degradation is not convincing, as MiDAC has multiple subunits. The result observed may be a biased outcome from DNTTIP1 degradation. To further confirm the observation is really coming from MiDAC malfunction, the same experiments should be performed following depletion of other subunit(s), or through a mutation of MiDAC catalytic activity.

The MiDAC complex has previously been shown to be dependent on DNTTIP1 (Turnbull et al. 2020 and Mondal et al. 2020, and Wang et al. 2022). We suggest that the reciprocal gene expression effects convincingly demonstrate Y654S creates a hyperactive MiDAC complex. Mutation to reduce the catalytic activity of HDAC1 is an interesting idea, but any effects would likely be obscured since HDAC1 is far more abundant in other HDAC complexes (NuRD, CoREST, SIN3A/B etc).

6) The connection between MiDAC hyperactivity and MAP2K6 upregulation is intriguing but needs more validations. Is the p38 phosphorylation altered Y654S-expressing cells? Does p38 inhibition rescue any of the cellular phenotypes observed (e.g., proliferation defects, cytokine secretion)?

This is an interesting point. Given that MAP2K6 shows very low expression, and the hyperactive complex would increase the inhibition, we do not expect a substantial effect of Y654S on MAP2K6 under normal conditions. Rather, Y654S would suppress MAP2K6 under p38 activating conditions, such as cellular stresses. We have added this point in the discussion.

To experimentally address this aspect would require setting up new models; genetically engineered cells with the Y654S variant that are responsive to p38 stimulation. We would argue that such an investigation goes beyond the scope of this study.

Specific points:

1) Figure 5c: The labels “control r” and “control q” need clarification.

We have updated the figure legend to clarify these labels.

2) Figure 5e: The meaning of the color coding should be explained clearly in the legend.

Figure now includes a scale to the heat map data.

3) Figure 5d: To validate successful fractionation of cytosolic and nuclear components, the western blot should be presented as a single, horizontal blot. This would clearly demonstrate that nuclear markers are absent from the cytosolic fraction and vice versa.

We have now repeated the blot in duplicate (figure 5d). This shows rapid depletion of DNTTIP1 and a corresponding increase in MAP2K6 protein levels (Figure 5d). These data confirm that the increase in gene expression is reflected in protein levels.

4) Figure 4c, the Y and X-axis labels are too small to evaluate.

This has been relabelled.

5) Line 259 – Interpretation Issue: The conclusion that “the variant (which variant?) leads to a hyperactive MiDAC” is not well-supported. Which variant should be clearly specified, and the logic connecting the data to this conclusion must be more rigorously explained.

We have clarified this sentence and have specified the Y654S variant.

6) Line 111: Full name of gnomAD?

Genome Aggregation Database (gnomAD) has now been added

7) Line 141: is the clinical RNAseq data publicly available or generated by the author? Please specify.

This has now been specified in the data availability statement.

8) line 214: Fig 4e was referred to before Fig 4c and d. The organization of the figure should be changed.

We have changed the order of panels in the figure to match the text.

9) The discussion of auto-inhibitory loops in other HDAC complexes (SIN3B, Rpd3L) is interesting but superficial. The authors could explore more by comparing the MIDEAS loop with known regulatory mechanisms in other HDAC complexes.

Potential regulatory loops have only recently been seen in the cryoEM structures of the human SIN3B and the yeast SIN3S and one of the active sites is sterically hindered in the yeast SIN3L. In each publication the authors suggest that deeper mechanistic studies will be required to understand the role of these loops in substrate specificity and HDAC activity. We have adjusted the text to clarify.

10) All abbreviations should be defined upon their first appearance in the text.

We have done this

Reviewer #4 (Remarks to the Author):
